# The atypical chemokine receptor 3 interacts with Connexin 43 inhibiting astrocytic gap junctional intercellular communication

Amos Fumagalli[1], Joyce Heuninck[1], Anne Pizzoccaro[1], Enora Moutin [1], Joyce Koenen [2,3], Martial Séveno [4], Thierry Durroux[1], Marie-Pierre Junier[5], Géraldine Schlecht-Louf [2], Francoise Bachelerie[2], Dagmar Schütz[6], Ralf Stumm [6], Martine J. Smit[3], Nathalie C. Guérineau[1], Séverine Chaumont-Dubel [1] & Philippe Marin [1]✉

The atypical chemokine receptor 3 (ACKR3) plays a pivotal role in directing the migration of various cellular populations and its over-expression in tumors promotes cell proliferation and invasiveness. The intracellular signaling pathways transducing ACKR3-dependent effects remain poorly characterized, an issue we addressed by identifying the interactome of ACKR3. Here, we report that recombinant ACKR3 expressed in HEK293T cells recruits the gap junction protein Connexin 43 (Cx43). Cx43 and ACKR3 are co-expressed in mouse brain astrocytes and human glioblastoma cells and form a complex in embryonic mouse brain. Functional in vitro studies show enhanced ACKR3 interaction with Cx43 upon ACKR3 agonist stimulation. Furthermore, ACKR3 activation promotes β-arrestin2- and dynamin-dependent Cx43 internalization to inhibit gap junctional intercellular communication in primary astrocytes. These results demonstrate a functional link between ACKR3 and gap junctions that might be of pathophysiological relevance.

[1] Institut de Génomique Fonctionnelle, Université de Montpellier, CNRS, INSERM, Montpellier, France. [2] Université Paris-Saclay, Inserm, Inflammation, Microbiome and Immunosurveillance, 92140 Clamart, France. [3] Amsterdam Institute for Molecules Medicines and Systems, Division of Medicinal Chemistry, Faculty of Sciences, VU University Amsterdam, 1081 HV Amsterdam, The Netherlands. [4] Biocampus Montpellier, Université de Montpellier, CNRS, INSERM, Montpellier, France. [5] CNRS UMR8246, Inserm U1130, Neuroscience Paris Seine-IBPS, Sorbonne Universités, Paris, France. [6] Institute of Pharmacology and Toxicology, Jena University Hospital, 07747 Jena, Germany. ✉email: philippe.marin@igf.cnrs.fr

The atypical chemokine receptor 3 (ACKR3, previously called CXCR7) is a seven-trans-membrane (7-TM) receptor belonging to the CXC chemokine receptor family that binds to CXCL12 and CXCL11. ACKR3 is expressed in various tissues such as the heart[1], kidney[2], and brain[3]. In the last, ACKR3 is found in neurons, astrocytes, and vascular cells[3] and plays a pivotal role in interneuron migration[4,5] and leukocyte entry into the brain parenchyma[6]. ACKR3 is over-expressed in numerous cancer types[7], including glioma, where both its expression and activation have been positively correlated with increased proliferative state and invasiveness[8].

Despite this accumulating knowledge, the intracellular pathways underlying ACKR3-dependent effects remain poorly characterized. Several reports suggest that ACKR3 operates as a molecular chemokine sink[4,9,10] lowering the extracellular chemokine concentration without activating any intracellular pathway. Several lines of evidence indicate that ACKR3 does not couple to and activate G proteins but only engages β-arrestin-dependent pathways[11–13]. A single study suggests that some ACKR3 effects are mediated by G proteins in glial cells[14].

Over the last two decades, it has become evident that 7-TM receptors interact with large networks of proteins that finely control their targeting to specific subcellular compartments, their trafficking in and out of the plasma membrane and the nature of receptor-operated signal transduction[15]. In line with these findings, studies have shown that ACKR3 exerts its biological effects by interacting with diverse membrane receptors[16]. For example, ACKR3 might indirectly influence Gα$_i$ activation via its interaction with the other CXCL12 receptor CXCR4[13]. ACKR3 also associates with the epidermal growth factor receptor[17] to promote proliferation of tumor cells in an agonist-independent manner. More recently, ACKR3 was found to interact with the G protein-coupled receptor kinase 2 that is involved in ACKR3 endocytosis[9]. These results suggest a strong influence of ACKR3-interacting proteins in ACKR3 pathophysiological functions. They provided the impetus for the extensive characterization of the ACKR3 interactome, thanks to a proteomic strategy combining affinity purification of receptor-interacting proteins and their identification by high-resolution mass spectrometry.

This interactomic screen identified the gap junction protein Connexin 43 (Cx43) as an ACKR3-interacting protein. In the brain, Cx43 is mainly expressed in astrocytes where Cx43 channels are involved in several important physiological processes including glucose diffusion[18] and propagation of Ca$^{2+}$ waves[19]. Several studies have demonstrated that Cx43-mediated gap junctional intercellular communication (GJIC) is finely controlled by astrocytic receptors, especially 7-TM receptors[20]. Furthermore, both ACKR3 and Cx43 have been involved in glioma progression. While ACKR3 was found to be upregulated in glioma, promoting tumor cell proliferation, angiogenesis and resistance to chemotherapy[8], suppression of Cx43-dependent communication in glioma cells has been correlated with enhanced proliferation and invasiveness[21,22]. On the other hand, Cx43 was found to promote glioma cell migration and resistance to apoptosis, suggesting opposing roles of Cx43 in glioma progression[21,22]. However, the functional link between ACKR3 and GJIC remains unexplored.

In light of these findings, we explored whether ACKR3 controls Cx43-mediated GJIC in primary astrocytes by combining dye diffusion experiments with electrophysiological recordings of junctional currents. We show that agonist stimulation of ACKR3 substantially reduces GJIC in astrocytes through a mechanism involving Gα$_{i/o}$ proteins and β-arrestin2-dependent Cx43 internalization. These findings provide evidence of a physical and functional interaction between ACKR3 and Cx43 that might underlie their influence on the pathogenesis of glioma. They shed light on the key role played by ACKR3-interacting proteins in determining the functional outcome of this chemokine receptor upregulated and mobilized in a broad range of pathologic conditions such as cancers.

## Results

**ACKR3 interacts with the gap junction protein Cx43.** Due to the lack of an ACKR3 antibody providing receptor immunoprecipitation yields compatible with mass spectrometry analysis, we expressed hemagglutinin (HA)-tagged ACKR3 in human embryonic kidney (HEK293T) cells. ACKR3-interacting proteins were immunoprecipitated using an anti-HA monoclonal antibody immobilized onto agarose beads. Control immunoprecipitations were performed using cells transfected with an empty plasmid. Only proteins identified in all three biological replicates in at least one group were considered for further analysis. Label-free quantification (LFQ) of the relative protein abundances in immunoprecipitates obtained from cells expressing ACKR3 and control cells showed that 151 proteins were significantly more abundant in immunoprecipitates from ACKR3-expressing cells, than in immunoprecipitates from control cells (Supplementary Data 1). As expected, ACKR3 (bait protein) was the most enriched one (Fig. 1a). The ACKR3 interactome also included Clathrin (CLTC) and accessory proteins of the Rab5 and Rab3 complexes involved in ACKR3 internalization[23]. In addition, we identified several E3 ubiquitin-protein ligases (HUWE1, HECTD1, and AMFR), which might be responsible for basal ubiquitination of the receptor[12]. In line with previous findings indicating that ACKR3 promotes ERK1/2 phosphorylation via activation of MAP2K2[24], MAP2K2 was retrieved in our interactomic screen. Consistent with a previous study showing a constitutive interaction between ACKR3 and G proteins[13], we also identified Gα$_{i3}$ (GNAI3) as a putative ACKR3 partner.

The ACKR3 interactome also included the Gap Junction Alpha-1 protein (GJA1, also called Connexin 43, Cx43) as well as additional proteins previously shown to be physically or functionally linked to Cx43 and to regulate Cx43-mediated GJIC. These included (i) Dynactin 1 (DCTN1)[25], which is known to affect the cellular localization of Cx43; (ii) Ubiquilin-4 (UBQLN4)[26], which interacts with and promotes the degradation of Cx43; (iii) Cytochrome P450 oxidoreductase (POR)[27], whose downregulation triggers transcriptional repression and inhibition of Cx43; (iv) Solute carrier family 1 member 5 (SLC1A5)[28], which interacts with Cx43 to stimulate cytotrophoblast fusion; (v) the beta-subunit of the electron-transfer protein (ETFB)[29], which interacts with Cx43 to regulate mitochondrial respiration and reactive oxygen species signaling; and vi) the tubulin beta-3 chain (TUBB3, Fig. 1a), suggesting that Cx43 might regulate microtubule stability in contacting cells[30].

Given the recruitment of Cx43-associated proteins by ACKR3 (Supplementary Data 1) and the co-expression of ACKR3 and Cx43 in GFAP-positive astrocytes of the subventricular zone and surrounding blood vessels in several brain regions such as the striatum and the hippocampus (Fig. 1b), we next sought to validate ACKR3-Cx43 interaction in the brain. As ACKR3 exhibits a peak of expression in the brain at the embryonic stage[3], we immunoprecipitated HA-ACKR3 from embryonic brains of HA-ACKR3 knock-in mice[9]. Cx43 co-immunoprecipitation with the receptor (Fig. 1c) indicates that both proteins form a complex in vivo.

We next assessed ACKR3-Cx43 interaction in living cells using bioluminescence resonance energy transfer (BRET) (Fig. 1d). We monitored the BRET signal variation as a function of the Cx43-YFP expression under conditions of constant ACKR3-NLuc or CXCR4-NLuc expression. BRET variation was significantly better fitted with a one-site saturation curve than a line through origin

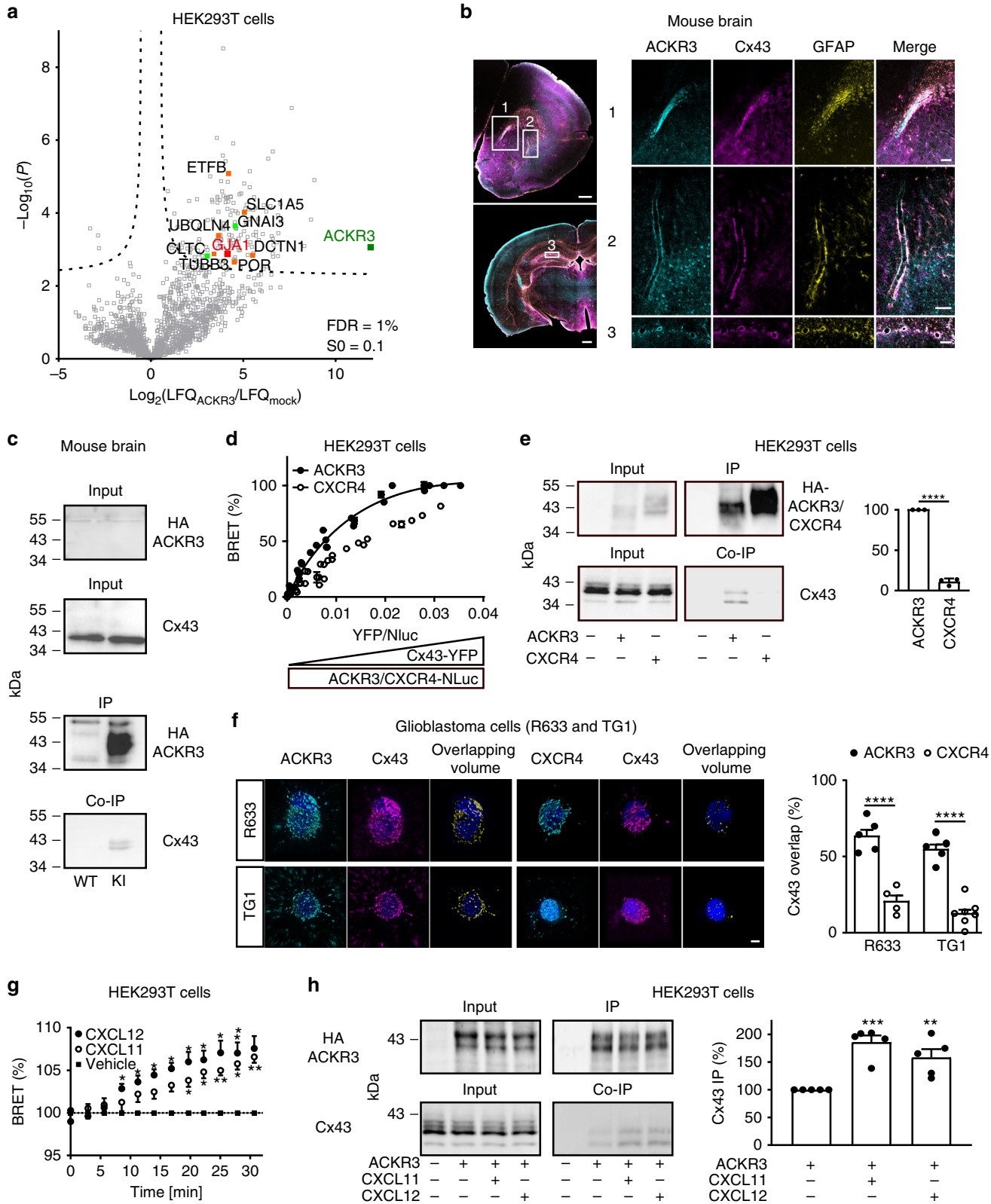

$(P < 0.0001)$ only for ACKR3 (Fig. 1d), arguing for a specific association of Cx43 with ACKR3 in living cells. Consistent with these findings, negligible Cx43 amounts co-immunoprecipitated with HA-CXCR4 in HEK293T cells, compared with HA-ACKR3, further supporting the specificity of ACKR3-Cx43 interaction (Fig. 1e). Double immunostaining of Cx43 and either ACKR3 or CXCR4 in two patient-derived glioblastoma-initiating cell lines with endogenous expression of the three proteins further indicated that a higher fraction of Cx43 is co-localized with ACKR3, in comparison with CXCR4 (Fig. 1f and Supplementary Fig. 1).

Collectively, the aforementioned results suggest that Cx43 already interacts with ACKR3 in the absence of an agonist. BRET experiments in HEK293T cells also showed that treating cells with

**Fig. 1 ACKR3 physically interacts with ACKR3. a** Volcano plot representing proteins identified by nanoLC-MS/MS in immunoprecipitations from HEK293T cells transfected with HA-ACKR3 or empty plasmid. Proteins known to physically or functionally interact with Cx43 (GJA1) and ACKR3 are represented in orange and green, respectively. **b** Representative images of GFP (Ackr3), Cx43 and GFAP immunostaining in brain from 8-week-old Ackr3-EGFP BAC mice. Scale bars = 500 μm for mosaics and 100 μm for others. **c** Representative WB of HA-immunoprecipitations from WT or HA-ACKR3 knock-in (KI) embryonic mouse brains. **d** Normalized BRET values measured in HEK293T cells co-expressing the indicated proteins. ACKR3-Cx43 BRET signal variations were fitted with the one-site total binding equation and background constraint to constant value of 0 using Prism (Extra-sum-of Square, F-test, $P < 0.001$ vs. line through the origin, $F_{(2,114)}$). **e** Representative WB of HA-immunoprecipitations performed from HEK293T cells expressing HA-ACKR3 or HA-CXCR4. The histogram shows the immunoreactive signals (normalized to the value in HA-ACKR3-expressing cells) of Cx43 co-immunoprecipitated with HA-receptors (Two-tailed unpaired $t$ test, $F_{(2,2)}$). **f** 3D reconstruction of ACKR3, CXCR4, and Cx43 immunostainings in TG1 and R633 cells. The overlapping volume represents the volume where each receptor is co-expressed with Cx43 (see Supplementary Fig. 1 for original immunoreactive signals). The histogram illustrates the percentage of Cx43 immunostaining in the overlapping volume (Two-way ANOVA, Bonferroni post-hoc, $F_{(1,17)}$). Scale bar = 3 μm. **g** Kinetics representing normalized BRET values measured in HEK293T cells co-expressing Cx43-YFP and ACKR3-NLuc challenged with either vehicle, 10 nM CXCL12 or 100 nM CXCL11. (Two-way ANOVA, Bonferroni post-hoc, $F_{(22,99)}$). **h** HEK293T cells expressing HA-ACKR3 challenged with vehicle, CXCL12 (10 nM) or CXCL11 (100 nM) for 30 min. Representative WB of HA-immunoprecipitations are illustrated. The histogram shows the immunoreactive signals (normalized to values from cells challenged with vehicle) of Cx43 co-immunoprecipitated with HA-ACKR3 (One-way ANOVA, Bonferroni post-hoc, $F_{(2,12)}$). All data are represented as means ± SEM. See the Statistics and Reproducibility section for the number of repetitions, exact $P$ values and symbol (*) legend. Source data and uncropped Western blots are provided as a Source Data file.

CXCL12 (10 nM) or CXCL11 (100 nM) time-dependently enhanced the interaction of ACKR3 with Cx43 (Fig. 1g). Co-immunoprecipitation of Cx43 with ACKR3 was also incrementally increased upon ACKR3 stimulation by CXCL12 or CXCL11 for 30 min (Fig. 1h), suggesting that the Cx43-ACKR3 interaction is a dynamic process depending on receptor activation.

**ACKR3 stimulation inhibits Cx43-mediated GJIC in astrocytes**. Given the co-expression of ACKR3 and Cx43 in astrocytes (Fig. 1b), we next investigated the effect of ACKR3 activation on GJIC in primary cultures of astrocytes. We first assessed GJIC by scrape loading followed by measuring the diffusion of the fluorescent dye Lucifer yellow (LY) from the scrape through the astrocytic syncytium[31]. As previously described[32] and in line with a high Cx43 expression[33], LY showed a large diffusion in untreated astrocytes, arguing for a robust GJIC (Fig. 2). Further supporting astrocyte coupling through gap junctions, treating cells with the gap junction inhibitor carbenoxolone (CBX, 50 μM) strongly inhibited LY diffusion (53 ± 6.5% inhibition vs. vehicle-treated cells, $p < 0.0001$, $n = 3$). Cell exposure to CXCL12 (10 nM) for 30 min also inhibited LY diffusion (47 ± 6.5% inhibition vs. vehicle-treated cells, $p < 0.0001$, $n = 3$). A similar level of inhibition (37 ± 6.5% inhibition vs. vehicle-treated cells, $p < 0.0001$, $n = 3$) was reached following exposure of astrocytes to CXCL11 (100 nM) for 30 min. In contrast, 5-min treatments with either CXCL12 or CXCL11 did not affect LY diffusion (Supplementary Fig. 2). CXCL12 is known to bind to CXCR4 in addition to ACKR3, while CXCL11 is also an agonist of CXCR3, another chemokine receptor subtype. Pre-treatment of cells with the CXCR4 antagonist AMD3100 (1 μM, 30 min), which alone did not change LY diffusion, did not prevent CXCL12-induced inhibition of GJIC. This suggests that the CXCL12 effect is mediated by ACKR3 and not CXCR4 (Fig. 2). Likewise, blocking CXCR3 by its antagonist NBI-74330 (1 μM, 30 min), which alone did not affect GJIC, did not prevent the ability of CXCL11 to inhibit GJIC. This result indicates that the CXCL11 effect is mediated by ACKR3.

We next performed double patch-clamp recordings of paired astrocytes in secondary culture to get more direct evidence that ACKR3 stimulation inhibits the electrical coupling between astrocytes.

The junctional current (Ij) was continuously monitored in cell pairs voltage-clamped at −50 mV. Ij was recorded for a 4-min control period before challenging cells with either the chemokines or CBX (Fig. 3a, b). While Ij magnitude remained stable during the 15-min recording of vehicle-treated astrocytes (+4.6 ± 20.8%,

$p = 0.9997$, $n = 6$), it strongly decreased as soon as 4.5 min after the onset of bath-application of CXCL12 or CXCL11, dropping down to 56.5 ± 16.7% ($p = 0.0194$, $n = 6$) and 52.0 ± 18.3% ($p = 0.0093$, $n = 6$), respectively, and remained inhibited as long as 10 min after the onset of treatments (Fig. 3b, c). As expected for a gap junction-mediated electrical coupling, CBX (100 μM) also drastically reduced Ij magnitude (51.9 ± 17.8%, $p = 0.0091$, $n = 6$). Altogether, these findings indicate that agonist ACKR3 stimulation inhibits GJIC in primary mouse astrocytes.

**ACKR3 inhibits GJIC through G proteins in astrocytes**. We next examined the possibility that G proteins could be involved in ACKR3-mediated GJIC inhibition, in light of our interactomic screen, which revealed constitutive association of ACKR3 with the $G\alpha_{i3}$ protein (Fig. 1a and Supplementary Data 1), and of previous findings suggesting that ACKR3 might activate $G_{i/o}$ proteins in primary rodent astrocytes[14]. Overnight pre-treatment of primary cultures of astrocytes with *pertussis toxin* (PTX, 100 ng/mL) abolished CXCL12- and CXCL11-induced inhibition of GJIC, whereas it did not reverse CBX-induced GJIC inhibition (44.6 ± 6.5% inhibition vs. vehicle, $p < 0.0001$, $n = 3$, Fig. 2). Interestingly, PTX significantly increased basal GJIC communication in astrocytes (130 ± 6.5% increase vs. vehicle, $p = 0.007$, $n = 3$, Fig. 2). Further confirming functional coupling between ACKR3 and $G\alpha_i$ proteins in astrocytes, treating cells with CXCL12 (10 nM) or CXCL11 (100 nM) for 5 min inhibited forskolin (10 μM)-induced cAMP production (40.8 ± 6.5, $p < 0.0001$ and 20.7 ± 6.5% inhibition, $p = 0.0254$ in cells exposed to CXCL12 and CXCL11, respectively, $n = 3$). Pre-treatment of cells with the CXCR4 antagonist AMD3100 (1 μM) for 30 min did not prevent CXCL12-induced inhibition of cAMP production (44.9 ± 6.5% vs. control, $p < 0.0001$ $n = 3$), indicating involvement of ACKR3 (and not CXCR4) in the CXCL12 effect (Supplementary Fig. 3a). As expected, and reminiscent of GJIC measurements, pre-treating astrocytes with PTX prevented CXCL12- and CXCL11-induced inhibition of cAMP production (Supplementary Fig. 3a).

These results contrast with previous findings indicating that ACKR3 is not able to activate G proteins in other cellular contexts[11,12,34], such as in HEK293 cells[13]. Corroborating the results of our interactomic screen performed in HEK293T cells (Fig. 1a), saturation BRET analysis confirmed that ACKR3 does interact with $G\alpha_{i3}$ proteins (Supplementary Fig. 3b). However, CXCL12 (10 nM) did not affect forskolin-induced cAMP production in HEK293T cells expressing human ACKR3 (Supplementary Fig. 3c), while it did inhibit cAMP production

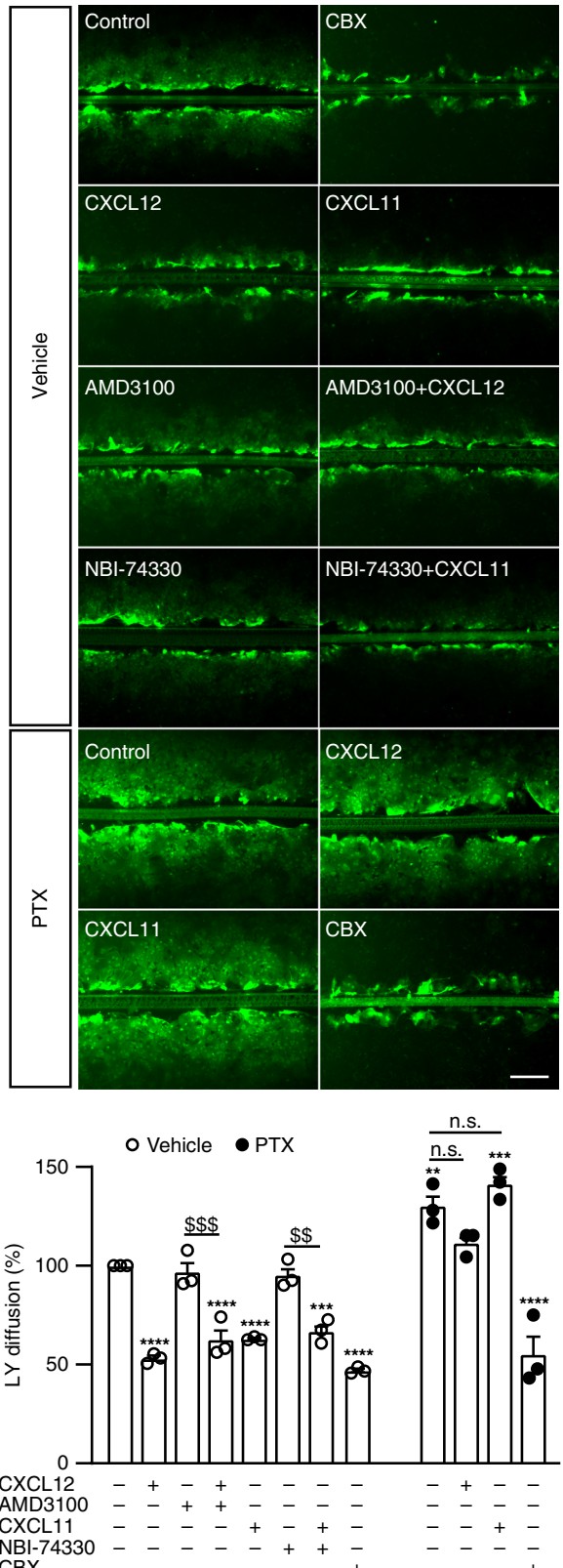

**Fig. 2 ACKR3 stimulation inhibits intercellular dye coupling in astrocytes.** Representative photomicrographs of scrape loading performed in confluent primary astrocyte cultures, taken ten minutes after the scrape in the presence of Lucifer yellow (1 mg/mL). All treatments (CXCL12 (10 nM), AMD3100 (1 μM), CXCL11 (100 nM), NBI-74330 (1 μM)) were applied for 30 min except for CBX (50 μM) and PTX (100 ng/mL) that were applied overnight. Scale bar = 200 μm. LY diffusion was calculated by measuring the distance from the scrape where LY fluorescence intensity is 50% of the maximal fluorescence. Values were normalized to LY diffusion in vehicle-treated astrocytes. Results represent the means ± SEM. One-way ANOVA was used for comparison within the vehicle-treated group $F_{(11,24)}$, whereas the two-way ANOVA was used for the comparison across vehicle and PTX groups $F_{(3,16)}$. For both Bonferroni post-hoc was used. See the Statistics and Reproducibility section for the number of repetitions, exact $P$ values and symbol ($^*$,$^\$$) legend. Source data are provided as a Source Data file.

## ACKR3 stimulation induces Cx43 internalization.

Consistent with already published results[9,35], ACKR3 was constitutively internalized in HEK293T cells in much higher amounts than CXCR4 (Supplementary Fig. 4a). Agonist stimulation of ACKR3 further increased the rate of receptor internalization (Supplementary Fig. 4b). As Cx43 activity is often regulated by the alteration of its trafficking[36], we next investigated if ACKR3-induced inhibition of GJIC in primary cultured astrocytes is mediated by Cx43 internalization. Cx43 immunostaining of vehicle-treated astrocyte cultures showed that Cx43 is primarily localized at the interface of cell–cell contacts, where Cx43 is organized as clusters (gap junctional plaques). This results in a typical "pavement-like" staining, but a little proportion of Cx43 was also detected intracellularly (Fig. 4a). In contrast, Cx43 was mainly detected in intracellular compartments in cultures exposed to CXCL12 (10 nM) or CXCL11 (100 nM). Furthermore, treatment of astrocytes with the dynamin inhibitor dynasore (80 μM) prevented CXCL12 and CXCL11 ability to promote Cx43 internalization (Fig. 4a). Dynasore also prevented the inhibition of GJIC induced by CXCL12 and CXCL11 treatments (Fig. 4b). In contrast, CBX was still able to decrease GJIC in dynasore-treated cells (57.0 ± 8.8% inhibition, $p < 0.0001$, $n = 3$, Fig. 4b). Collectively, these results suggest that ACKR3 stimulation promotes Cx43 internalization and inhibits GJIC through a dynamin-dependent mechanism in astrocytes.

To get more direct evidence that ACKR3 stimulation promotes Cx43 internalization, we performed time-lapse recordings of Cx43-GFP in living HEK293T cells. Treatment of cells co-expressing RedCherry-ACKR3 and Cx43-GFP with CXCL12 (10 nM) or CXCL11 (100 nM) removed Cx43 plaques localized at cell-cell contacts, whereas plaques were not affected in cells that do not express ACKR3 (Fig. 5 and Supplementary Fig. 5). Further supporting the internalization of Cx43 plaques upon ACKR3 stimulation, the formation of Cx43-containing buds was transiently observed prior to the removal of plaques (Fig. 5). Furthermore, removal of Cx43 plaques often started from their central part (Supplementary Fig. 5), consistent with previous findings indicating that Cx43 internalization primarily occurs from the center of the plaques[37].

We next explored whether ACKR3-induced Cx43 internalization modifies the subcellular localization of Cx30 in astrocytes, another connexin isoform abundantly expressed in this cell type[38]. Contrasting with what was observed for Cx43, Cx30 was mainly detected intracellularly in untreated astrocytes and exposure to CXCL12 or CXCL11 did not affect its intracellular localization (Supplementary Fig. 6, left panel). Treatment of astrocytes with dynasore (80 μM) induced a redistribution of Cx30 to the cell surface in cells exposed or not to CXCL11 and CXCL12

in cells expressing CXCR4 (43.9 ± 10.5% inhibition, $p = 0.0018$, $n = 3$). Likewise, activation of mouse recombinant CXCR4 by CXCL12, but not mouse ACKR3, caused a decrease in the BRET signal between venus-G$\gamma_2$ and G$\alpha_{i3}$-RLuc in HEK293T cells (Supplementary Fig. 3d), suggesting that the ability of ACKR3 to activate G$\alpha_{i/o}$ proteins in mouse astrocyte cultures does not reflect different coupling properties of mouse and human ACKR3.

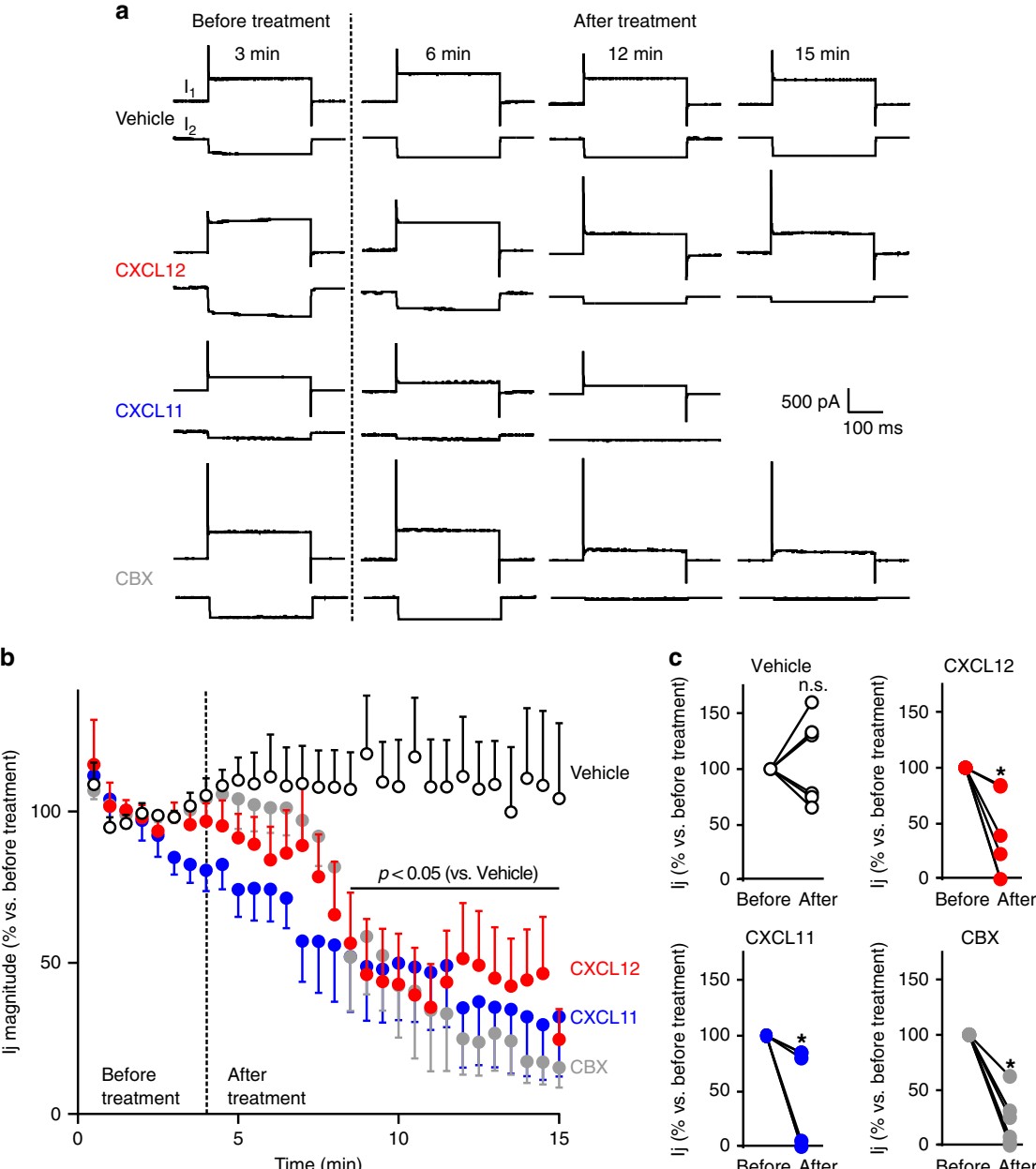

**Fig. 3 ACKR3 stimulation impairs gap junctional electrical coupling between astrocytes. a** Representative membrane currents recorded in cell pairs of secondary astrocyte cultures (whole-cell configuration of the patch-clamp technique, holding potential −50 mV). Each chart recording illustrates the current recorded in the stimulated cell (I1, upper trace, in response to depolarizing voltage steps, 40 mV amplitude, 300 ms duration, every 30 s) and the junctional current recorded in the non-stimulated cell (I2, bottom trace). I1 and I2 were continuously monitored before and after the treatment of cells with either vehicle, CXCL12 (10 nM), CXCL11 (100 nM), or CBX (100 μM). **b** Time-course of the effect of CXCL12, CXCL11, and CBX on Ij magnitude. Perfusion of the recorded cell pairs with the different compounds started at the time indicated by the dotted line. Data are expressed as a percentage of Ij (normalized to the average ratio recorded during the first 4 min before any treatment). Values are means ± SEM (Two-way ANOVA, Bonferroni post-hoc, $F_{(87,573)}$). **c** Ij before and after treatment for each cell pair recorded in the different conditions are plotted. Data are expressed as a percentage of Ij (normalized to the average ratio recorded during the first 4 min before any treatment, Two-tailed Wilcoxon test). See the Statistics and Reproducibility section for the number of repetitions, exact $P$ values and symbol (*) legend. Source data are provided as a Source Data file.

(Supplementary Fig. 6, right panel). Collectively, these results suggest that Cx30 is "constitutively" internalized in astrocytes through a mechanism independent of ACKR3 stimulation and that ACKR3-induced internalization of Cx43 does not influence Cx30 subcellular localization.

**ACKR3-mediated GJIC inhibition depends on β-arrestin2.**
ACKR3 is known to signal through β-arrestin2, which has also

been involved in its internalization[12,35]. Corroborating these observations, BRET experiments showed a constitutive interaction between ACKR3 and β-arrestin2 in HEK293T cells (Supplementary Fig. 4c) that increased in a concentration-dependent manner upon CXCL12 and CXCL11 challenge (Supplementary Fig. 4d). In contrast, no constitutive association of CXCR4 with β-arrestin2 was observed (Supplementary Fig. 4c). Both constitutive and agonist-induced interaction of ACKR3 with β-arrestin2 was confirmed by co-immunoprecipitation experiments in

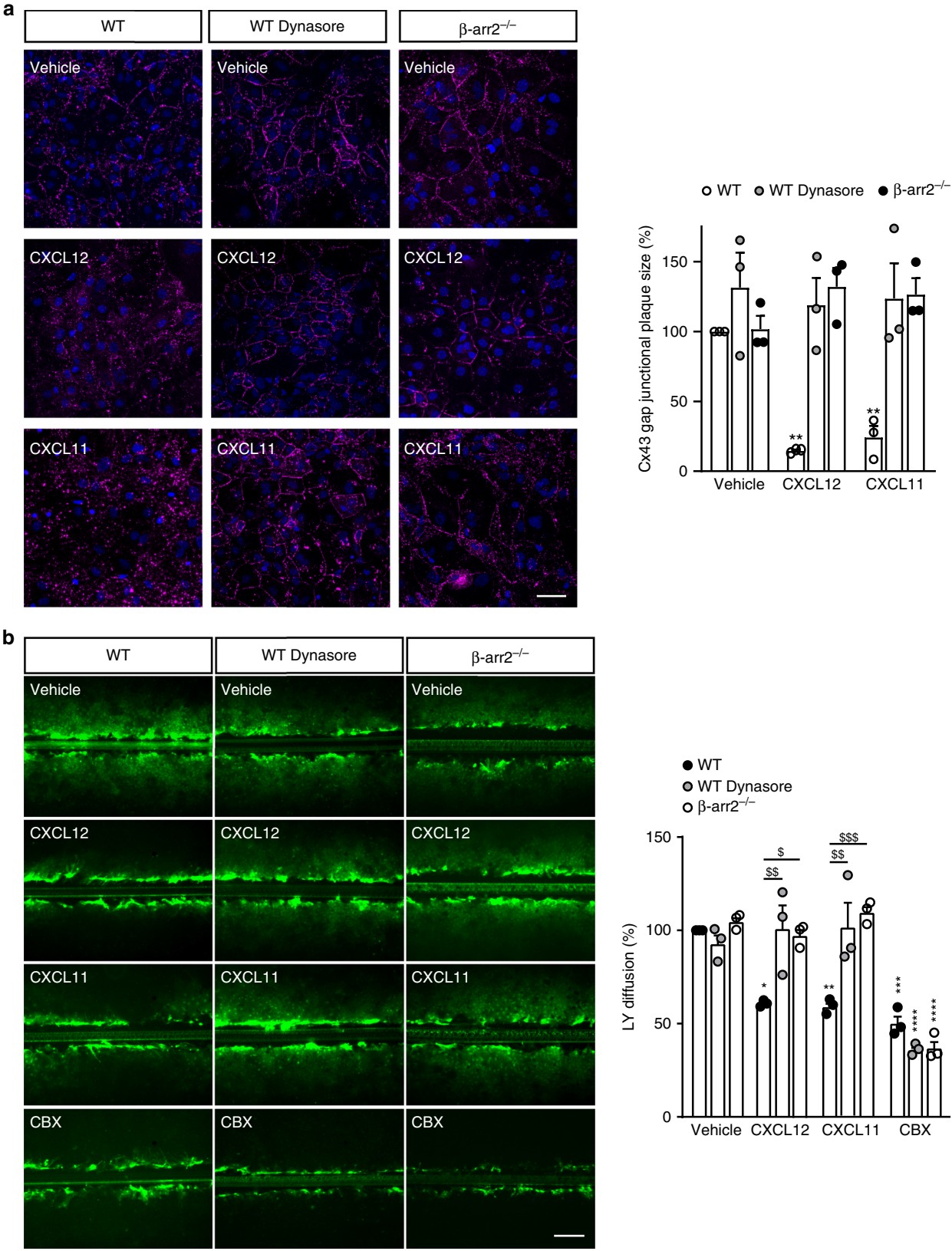

HEK293T cells co-expressing HA-tagged ACKR3 and FLAG-tagged β-arrestin2 (Supplementary Fig. 4e). Therefore, agonist stimulation of ACKR3 induces concomitant recruitment of β-arrestin2 and Cx43 (Supplementary Fig. 4e). However, no BRET signal was detected in HEK293T cells expressing YFP-tagged Cx43 and RLuc-tagged β-arrestin2 in the presence of ACKR3

with or without CXCL12 stimulation, thereby ruling out a direct interaction between Cx43 and β-arrestin2 (Supplementary Fig. 4f).

We next examined the impact of CXCL12 and CXCL11 on Cx43 subcellular localization in astrocytes from β-arrestin2 knock out (βarr2$^{-/-}$) mice. Contrasting to what was observed in

**Fig. 4 ACKR3 stimulation regulates Cx43 cellular localization and inhibits GJIC in astrocytes. a** Confocal images of Cx43 immunostaining of confluent primary astrocyte cultures from WT or β-arr2$^{-/-}$ mice representative of three independent experiments performed on different sets of cultured cells exposed for 30 min to either vehicle or CXCL12 (10 nM) or CXCL11 (100 nM) in the absence or presence of Dynasore (Dyn, 80 μM). Scale bar = 50 μm. The histograms show the size of Cx43 gap junctional plaques (clusters of channels present at the cell-cell interface). Data were normalized to the one measured in vehicle-treated cultures. They are the means ± SEM of values (Two-way ANOVA, Bonferroni post-hoc F(4,18)). **b** Representative photomicrographs of scrape loading in confluent primary astrocyte cultures obtained from embryonic WT and β-arr2$^{-/-}$ mice taken 10 min after the scrape in the presence of LY. CXCL12 (10 nM), CXCL11 (100 nM), and Dynasore (Dyn, 80 μM) were applied for 30 min. CBX (50 μM) was applied overnight. Scale bar = 200 μm. LY diffusion was calculated by measuring the distance from the scrape where LY fluorescence intensity is 50% of the maximal fluorescence. Values were normalized to LY diffusion in vehicle-treated astrocytes. Results represent the means ± SEM of values (Two-way ANOVA, Bonferroni post-hoc, F(6,24)). See the Statistics and Reproducibility section for the number of repetitions, exact *P* values and symbol (*/$) legend. Source data are provided as a Source Data file.

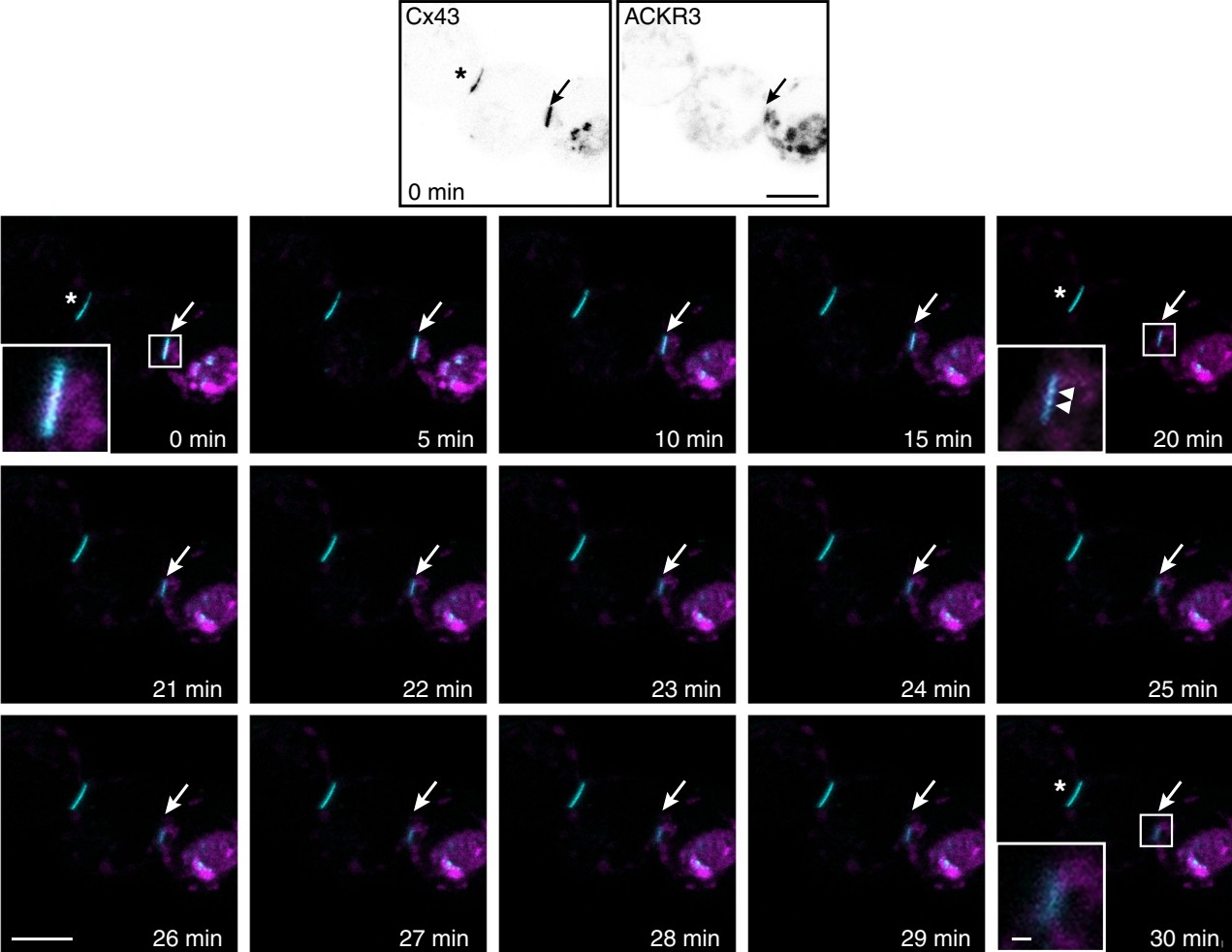

**Fig. 5 ACKR3 activation promotes Cx43 internalization.** Time-lapse images of gap junction plaques in HEK293T cells co-expressing Cx43-GFP and RedCherry-ACKR3 and exposed to CXCL12 (10 nM) for 30 min. The two upper images show GFP (Cx43, left panel) and RedCherry (ACKR3, right panel) fluorescence signals, respectively, before the onset of chemokine application. Images below show Cx43-GFP fluorescence at the times indicated. A zoomed image of a Cx43-GFP plaque is illustrated at 0, 20 and 30 min in the left bottom corner of the image. The arrows show the removal of the plaque and the arrowheads Cx43-GFP vesicle budding. The decrease in Cx43-GFP fluorescence was not due to photobleaching, as CXCL12 exposure did not induce the removal of a gap junction plaque located at the interface of cells that do not express ACKR3 (*). Scale bars = 10 and 1 μm for time-lapse and magnified pictures, respectively. See the Statistics and Reproducibility section for the number of repetitions.

astrocytes from wild type mice, exposing primary astrocyte cultures from βarr2$^{-/-}$ mice to CXCL12 (10 nM) or CXCL11 (100 nM) did not promote Cx43 internalization (Fig. 4a). Consistently, neither CXCL12 nor CXCL11 inhibited GJIC in astrocytes from these mice (Fig. 4b). In addition, the extent of LY diffusion through vehicle-treated astrocytes was similar to that measured in astrocytes from WT mice and CBX was still able to

inhibit astrocytic GJIC in βarr2$^{-/-}$ mice (67.8 ± 8.2% inhibition, *p* < 0.0001, *n* = 3, Fig. 4b).

## Discussion
ACKR3 is involved in key biological processes such as migration[10], proliferation[39], and differentiation[40,41] of different cell populations, but the cellular pathways transducing these effects

remained poorly understood. To address this issue, we characterized the ACKR3 interactome in HEK293T cells and identified Cx43 as an ACKR3-interacting protein. Intriguingly, ACKR3 was previously found to interact with the gap junction beta-2 protein GJB2 (or Cx26), but not Cx43, in a large interactomic screen exploring the interactome of 1125 GFP-tagged proteins expressed in HeLa cells[42]. These distinct results might reflect the different cell backgrounds (HEK293T vs. HeLa) used in both studies and suggest that ACKR3 physically associates with gap junctions composed of distinct connexin families in a cell type-dependent manner. The ability of ACKR3 and Cx43 to form a complex in vivo was further validated by co-IP in mouse brain, where both proteins are co-expressed in GFAP-positive astrocytes of the subventricular zone and surrounding blood vessels. ACKR3 is also present in neurons[3] that do not express Cx43[43], indicating that the interaction between ACKR3 and Cx43 only occurs in restricted brain cell populations. Whether ACKR3 modulates GJIC in other cell types expressing different connexin isoforms such as neurons remains to be explored. Moreover, using complementary approaches, we showed that Cx43 interacts with ACKR3 but not with the functionally related CXCR4 in both HEK293T cells and glioblastoma-initiating cell lines.

There is convergent evidence indicating that ACKR3 and Cx43 exert an opposite influence upon various physiological and pathological processes such as neuronal migration[4,5,44,45], leukocyte entry[6,46] into the brain and formation or progression of certain cancer types[8,47]. For instance, a large body of evidence indicates an increased expression of ACKR3, which favors tumor cell proliferation and invasion as well as angiogenesis[8], in various cancers such as glioma[48]. In contrast, studies support the notion that connexins, including Cx43, are tumor suppressors even though recent evidence suggest that Cx43 may also promote tumor cell migration and metastasis through both GJIC-dependent and independent pathways[47]. Likewise, ACKR3 expression is increased during experimental autoimmune encephalomyelitis (EAE), a preclinical model of multiple sclerosis (MS), which is ameliorated by treatment with an ACKR3 antagonist[6], whereas astrocytic Cx43 expression is decreased in MS and other inflammatory demyelinating brain disorders. Furthermore, Cx43 loss is mainly found in actively demyelinating and chronically active lesions and is associated with rapid disease progression[49]. Corroborating this opposite influence upon common pathophysiological processes, we show that agonist stimulation of ACKR3 inhibits GJIC in primary astrocyte cultures using fluorescent dye diffusion and whole-cell patch-clamp recordings. These observations provide evidence of a direct functional link between ACKR3 and gap junctions that might contribute to the role of ACKR3 in glioma progression. Reminiscent of the specific association of Cx43 with ACKR3 but not with CXCR4, only ACKR3 stimulation reduced GJIC in astrocytes. The role of the physical ACKR3-Cx43 interaction in receptor-operated inhibition of GJIC could not be fully established in the absence of the identification of binding motifs within the sequences of both partners.

In an effort to identify the signaling pathways involved in the ACKR3-mediated inhibition of GJIC, we showed that inhibiting $G\alpha_{i/o}$ protein activation by PTX treatment abolished the ACKR3 inhibitory effect in primary cultured astrocytes. Consistent with previous studies indicating that Cx43 is negatively regulated by the activation of numerous GPCRs[20], the increase in basal GJIC elicited by PTX might reflect a blockade of $G_{i/o}$-coupled receptors endogenously expressed in astrocytes. In untreated conditions, these GPCRs would inhibit Cx43-mediated GJIC as a result of their activation by gliotransmitters released by astrocytes or constitutive activity. The inhibitory effect of PTX upon ACKR3-mediated inhibition of GJIC was more unexpected given the current classification of ACKR3 as an atypical chemokine receptor unable to activate G proteins. However, it corroborates a previous report indicating that the receptor can signal through $G\alpha_{i/o}$ proteins in rodent astrocytes and human glioma cells[14]. The ability of ACKR3 to activate G proteins has so far only been demonstrated in glial cells, but not in other cell types like HEK293 cells[11–13,34]. Consistent with these findings, we showed that ACKR3 stimulation inhibits cAMP production through a $G\alpha_{i/o}$-dependent mechanism in primary astrocytes, but not in HEK293T cells. Furthermore, this different capability to activate $G_{i/o}$ proteins was not due to different coupling properties between human and mouse ACKR3, as stimulation of mouse ACKR3 expressed in HEK293T cells did not activate G proteins. Though one cannot entirely rule out that ACKR3 stimulation promotes the release of GPCR agonists by glial cells, such as adenosine and glutamate, which would in turn activate their respective endogenously expressed receptors to inhibit cAMP production, such an indirect activation of other $G\alpha_{i/o}$-coupled receptors is unlikely, due to the short chemokine challenging protocol used. Accordingly, the present findings together with previously published results suggest that astrocytes constitute a unique cellular environment where ACKR3 can activate G proteins. This unique capability of astrocytic ACKR3 to activate $G\alpha_{i/o}$ proteins might rely on the specific expression of not yet identified receptor partners or adaptor proteins.

Beyond G protein-dependent mechanisms, Cx43 activity is often regulated by the alteration of its trafficking[36]. Several protein kinases including some members of the MAP kinase pathways and Cx43 partners such as clathrin, myosin, actin, and drebrin, have been involved in Cx43 endocytosis[50–53]. Here, we provide convergent evidence indicating that GJIC inhibition elicited by ACKR3 stimulation in astrocytes is a consequence of internalization of Cx43 involved in gap junctions: (i) treatment of astrocyte cultures with either CXCL12 or CXCL11 triggered concomitant Cx43 internalization and GJIC inhibition; (ii) inhibition of Cx43 internalization by the dynamin inhibitor Dynasore prevented the ability of ACKR3 stimulation to inhibit GJIC; and (iii) GJIC inhibition and concomitant Cx43 internalization elicited by ACKR3 activation were abolished in astrocytes from β-arr2$^{-/-}$ mice, consistent with the ability of the receptor to interact with β-arrestin2[9]. Previous studies have demonstrated that recruitment of β-arrestin2 by ACKR3 depends on its C-terminal domain[34] and likely involves the phosphorylation of Ser/Thr clusters by protein kinase(s) such as GRK2 and GRK5[9]. Co-IP experiments indicated that ACKR3 forms a complex with both Cx43 and β-arrestin2 and that their recruitment is enhanced upon agonist receptor stimulation, suggesting that ACKR3 might concomitantly recruit Cx43 and β-arrestin2 via different binding motifs.

The prevailing view on GPCR endocytosis is that the phosphorylation of ligand-activated GPCRs by GRKs and other kinases induces recruitment of β-arrestins, which then promotes clathrin-dependent receptor endocytosis. In agreement with this model, both basal and agonist-induced ACKR3 internalization were initially reported to depend on β-arrestins[34,35]. These findings favor the possibility that Cx43 is directly co-internalized with activated ACKR3 within a complex comprising ACKR3, Cx43, and β-arrestin2. Nevertheless, the notion that ACKR3 requires β-arrestins to internalize and sequester chemokines has been challenged by recent reports indicating that β-arrestins are dispensable for ACKR3 endocytosis and ACKR3-mediated scavenging of CXCL12 in mammalian cell cultures[54], zebrafish[23], and mouse brain[9]. Accordingly, Cx43 internalization in CXCL12/CXCL11-exposed astrocytes might occur independently of ACKR3 internalization through an indirect, β-arrestin2-dependent, signaling mechanism.

Protein–protein interactions play a key role in numerous biological processes, and their dysfunctions have been involved in the pathogenesis of a range of human diseases, including cancer. Abnormal protein–protein interactions contribute to all phases of oncogenesis, including tumor cell proliferation and survival, inflammation, invasion, and metastasis. Identifying and characterizing the function of protein–protein interactions involving molecules upregulated and mobilized in a diversity of cancers, such as ACKR3, is thus crucial to understand mechanisms underlying cancer progression and develop new therapeutic strategies[55]. In this context, the current study provides evidence for a physical and functional link between the CXCL11/CXCL12/ACKR3 axis and Cx43, one of the members of the gap junction protein family that can act as tumor suppressor or promote tumor progression and metastasis, depending on the cancer type and cancer stage[47]. This interaction might not only contribute to the development of various cancers such as glioma, but also play a key role in physiological processes under the control of ACKR3 or depending on proper function of gap junction channels in astrocytes, such as interneuron migration[4,5], blood brain barrier permeability[6], and synaptic transmission and plasticity[56]. Finally, the present study provides insight into the persistent conundrum of the role of G proteins in ACKR3-mediated effects by confirming the previously described engagement of G protein-dependent signaling in astroglial cells[14], thereby suggesting a cell type-dependent involvement of G proteins in ACKR3-operated signaling and function.

## Materials and methods

**Materials**. HA-ACKR3 was obtained introducing a single HA-tag at the N-terminal by PCR. The construct was then cloned in the pCDNA3.1 vector using the BamHI and XbaI restriction enzymes. 3XHA-CXCR4-pCDNA3.1 construct was obtained from cDNA.org (Bloomsburg University). The pcDNA 3.1 plasmids encoding ACKR3-NLuc and CXCR4-NLuc were provided by Dr. M.J. Lohse (Max-Delbrück-Zentrum für Molekulare Medizin (MDC), Berlin, Germany). The Cx43-YFP-pCDNA3.1, Cx43-GFP-pEGFPN1 and RedCherry-ACKR3-pCDNA3.1 plasmids were obtained from the human ORFeome collection (IGMM, Montpellier, France). The FLAG-SNAP-ACKR3 and FLAG-SNAP-CXCR4 constructs (in pcDNA3.1) were provided by Cisbio Bioassays (Codolet, France). The ACKR3-YFP and the CXCR4-YFP constructs were described in[13] and[57], respectively. pcDNA3.1 plasmids encoding Venus-Tagged G protein subunit gamma 2 (Venus-γ2), FLAG-tagged G protein subunit beta 2 (FLAG-β2), β-arrestin2-RLuc and Gα$_{i3}$-RLuc were provided by Dr. D. Maurel (IGF, Montpellier, France). Human β-arrestin2-FLAG-pcDNA3.1 plasmid was provided by Dr. C. Mendre (IGF, Montpellier, France). Mouse ACKR3 and CXCR4 constructs were described in refs. [58,59], respectively.

EGFP-ACKR3 BAC mice were from GENSAT and characterized in ref. [2,4]. They were maintained heterozygous on Hsd:ICR (CD-1®) background. The HA-ACKR3 mouse models were designed and generated by GenOway (Lyon, France) and already characterized in[9]. C57BL/6J β-arr2$^{-/-}$ mice were generated from heterozygous β-arr2$^{+/-}$ mice generated in Prof J. Lefkowitz's laboratory (Duke University, USA). Mice were housed under standardized conditions with a 12-h light/dark cycle, stable temperature (22 ± 1 °C), controlled humidity (55 ± 10%), and free access to food and water. Experiments on animals conformed to European ethics standards (86/609-EEC) and to decrees of the French National Ethics Committee (N°87/848) for the care and use of laboratory animals. Protocols were approved by the University of Montpellier ethics committee for animal use (CEEA LR 34, #7251).

Recombinant mouse CXCL11 (Ref 572-MC), Human/Feline/Rhesus Macaque CXCL12 (Ref 350-NS) and NBI-74330 (Ref 4528/10) were purchased from R&D System, AMD3100 from Tocris (Ref 3299), Lucifer Yellow CH dilithium salt (LY, Ref L0259), Dynasore hydrate (Ref D7693), carbenoxolone disodium salt (CBX, Ref C4790) and pertussis toxin (Ref 516560) from Merck.

**Primary antibodies**. For the complete list of primary antibodies and related information, see Supplementary Table 1.

**Cell cultures and transfection**. HEK293T cells, purchased from ATCC (Anassas, VI, ATCC, CRL-1573), were grown in Dulbecco's Modified Eagle's Medium (DMEM, Thermo Fisher Scientific, Ref 419960) supplemented with 10% heat-inactivated fetal bovine serum (Thermo Fisher Scientific, Ref 10099-133) and maintained in humidified atmosphere containing 5% CO$_2$ at 37 °C. Cells were passed twice a week and used between passages 10 and 20. Absence of mycoplasma was assessed every month using the MycoAlert Mycoplasma Detection Kit (Lonza,

Ref LT07-118). HEK293T cells were transfected in suspension using poly-ethylenimine (PEI, Polyscience, Ref 24765). Cells were seeded one day prior transfection and used two days after transfection for all experiments except for BRET. For BRET cells were transfected in suspension 24 h before experiment. Mock cells were transfected with empty plasmids.

Primary and secondary cultures of astrocytes were prepared from wild type and β-arr2$^{-/-}$ mice embryos (embryonic day E16.5). Forebrains were dissected and cells were dissociated mechanically before seeding in culture medium containing a 1:1 mixture of DMEM and F-12 nutrient (Merck, Ref D057) supplemented with glucose (30 mM), glutamine (2 mM), NaHCO$_3$ (13 mM), HEPES buffer (5 mM, pH 7.4), penicillin-streptomycin (100 unit/mL-0.1 mg/mL, Thermo Fisher Ref 15140-122) and 10% heat-inactivated Nu-Serum (Corning, Ref 355500). The medium was changed one week after seeding and then twice a week. Five weeks after seeding, cultures were starved from serum in DMEM supplemented with penicillin-streptomycin (100 unit/mL–0.1 mg/mL) overnight before experiment. For secondary culture preparation, cells were washed with PBS and incubated for 15 min with trypsin-EDTA 0.05% (Thermo Fisher Scientific, Ref 25300-054) at 37 °C. Cells were then centrifuged at 200 × g for 5 min, suspended in complete DMEM/F-12 medium, plated on polyornithine-coated glass coverslips and processed for double patch-clamp recordings the following day. Primary cultures were used for all experiments but double patch-clamp recordings that were performed on secondary cultures.

Patient-derived cell lines TG1 and R633, isolated from neurosurgical biopsy samples of human glioblastoma and characterized for their stem-like and tumor-initiating properties, were cultured in Dulbecco's modified Eagle's:F-12 medium (1:1) containing the N2, G5 (containing FGF and EGF) and B27 supplements (all from Invitrogen, France)[60,61]. Samples were obtained with informed consent of patients to use them in research. The institutional review board of the Sainte-Anne Hospital Center - University Paris Descartes (Comité de protection des personnes Ile de France III) approved the study protocol (Protocol number DC-2008-323).

**Co-immunoprecipitation**. HEK293T cells were lysed in ice-cold lysis buffer containing 1% n-Dodecyl-β-D-Maltopyranoside (DDM, Antrace, Ref D310), 20 mM Tris-HCl (pH 7.5), 100 mM NaCl, 2.5 mM CaCl$_2$, phosphatase inhibitors (NaF, 10 mM; Na$^+$-vanadate, 2 mM; Na$^+$-pyrophosphate, 1 mM; and β-glycerophosphate, 50 mM) and the cOmplete Protease Inhibitor Cocktail (Merck, Ref 11836145001). Samples were maintained under gentle agitation for 1 h at 4 °C and centrifuged for 15 min at 15,000 × g to eliminate insoluble material. Soluble proteins were quantified by bicinchoninic acid assay (BCA, Merck, Refs B29643 and C2284) and equal protein amounts (5 or 1.5 mg for IP followed by MS/MS or WB, respectively) were incubated with agarose-conjugated anti-HA antibody (Merck, Ref A2095) overnight at 4 °C. Samples were then washed twice with an ice-cold solution of 0.5 M of NaCl and phosphatase inhibitors and three times with 150 mM NaCl and phosphatase inhibitors. Immunoprecipitated proteins were then eluted in Laemmli sample buffer.

Co-immunoprecipitation from embryonic brains (E17) was performed as described in[9]. Briefly, brains were isolated and snap frozen in liquid nitrogen. Two brains per conditions were pooled and sonicated in lysis buffer containing (150 mM NaCl, 5 mM EDTA, 50 mM Tris-HCl, pH 7.4 and DDM 1.5%) plus the cOmplete Protease Inhibitor Cocktail. Samples were maintained for 1 h at 4 °C under gentle agitation and centrifuged for 30 min at 21,000 × g. Soluble proteins were quantified by BCA and equal protein amount (5 mg) per each condition was incubated with the agarose-conjugated anti-HA magnetic beads (Thermo Fisher Scientific, Ref 88836) overnight at 4 °C under constant rotation. Immunoprecipitated proteins were then eluted in Laemmli sample buffer.

**Protein identification by mass spectrometry**. Immunoprecipitated proteins were separated by SDS-PAGE and stained with Protein Staining Solution (Euromedex, Ref 10-0911). Gel lanes were cut into seven gel pieces and destained with 50 mM TriEthylAmmonium BiCarbonate (TEABC, Merck, Ref T7408) followed by three washes in 100% acetonitrile. After reduction (with 10 mM dithiothreitol in 50 mM TEABC at 60 °C for 30 min) and alkylation (with 55 mM iodoacetamide TEABC at room temperature for 60 min), proteins were digested in-gel using trypsin (500 ng/band, Gold, Promega, Ref V5280)[62]. Digest products were dehydrated in a vacuum centrifuge and reduced to 3 μL. The generated peptides were analyzed online by nano-flow liquid chromatography coupled to tandem-mass spectrometry (nanoLC-MS/MS) using an Orbitrap Elite mass spectrometer (Thermo Fisher Scientific, Waltham USA) coupled to an Ultimate 3000 HPLC (Thermo Fisher Scientific). Desalting and pre-concentration of samples were performed on-line on a Pepmap® pre-column (0.3 mm × 10 mm, Dionex). A gradient consisting of 0–40% B for 60 min and 80% B for 15 min (A = 0.1% formic acid, 2% acetonitrile in water; B = 0.1% formic acid in acetonitrile) at 300 nL/min was used to elute peptides from the capillary reverse-phase column (0.075 mm × 150 mm, Acclaim Pepmap 100® C18, Thermo Fisher Scientific). Eluted peptides were electrosprayed online at a voltage of 1.8 kV into the Orbitrap Elite mass spectrometer. A cycle of one full-scan mass spectrum (MS1, 400–2,000 m/z) at a resolution of 120,000 (at 400 m/z), followed by 20 data-dependent tandem-mass (MS2) spectra was repeated continuously throughout the nanoLC separation. All MS2 spectra were recorded using normalized collision energy (33%, activation Q 0.25 and activation time 10 ms) with an isolation window of 2 m/z. Data were acquired using the Xcalibur software (v 2.2).

For all full scan measurements with the Orbitrap detector a lock-mass ion from ambient air (m/z 445.120024) was used as an internal calibrant[63]. Mass spectra were processed using the MaxQuant software package (v 1.5.5.1) and MS2 using the Andromeda search engine "[http://coxdocs.org/doku.php?id=maxquant:andromeda:start]" against the UniProtKB Reference proteome UP000005640 database for Homo sapiens (release 2017_10) and the contaminant database in MaxQuant. The following parameters were used: enzyme specificity set as Trypsin/P with a maximum of two missed cleavages, Oxidation (M) and Phosphorylation (STY) set as variable modifications and carbamidomethyl (C) as fixed modification, and a mass tolerance of 0.5 Da for fragment ions. The maximum false peptide and protein discovery rate was specified as 0.01. Seven amino acids were required as minimum peptide length. When precursor peptides were present in MS1 spectra but not selected for fragmentation and identification by MS2 in given runs, peptide identification were based on accurate mass and retention times across LC-MS runs using the matching between runs tool in MaxQuant. Only proteins identified in all three biological replicates in at least one group were considered for further analysis. Relative protein quantification in IP from ACKR3-expressing cells and Mock cells was performed using the label-free quantification (LFQ) algorithm "[https://maxquant.net/maxquant/]". For statistical analysis, missing values were defined using the imputation tool of the Perseus software (v. 1.5.6.072) "[https://maxquant.net/perseus/]".

**Western blotting**. Proteins were separated by SDS-PAGE onto 10% polyacrylamide gels and transferred electrophoretically to nitrocellulose membranes (Bio-Rad, Ref 1704271). Membranes were incubated in blocking buffer (Tris-HCl, 50 mM, pH 7.5; NaCl, 200 mM; Tween-20, 0.1% and skimmed dried milk, 5%) for 1 h at room temperature and overnight with primary antibodies in incubating buffer (Tris-HCl, 50 mM, pH 7.5; NaCl, 200 mM; Tween-20, 0.1% and Bovine Serum Albumin (Merck, Ref A2153), 5%) at 4 °C. Then membranes were immunoblotted with either anti-mouse (Merck, Ref GENA931V) or anti-rat (Jackson ImmunoResearch, Ref 112-035-003) or anti-rabbit (Merck, Ref GENA934V) horseradish peroxidase (HRP)-conjugated secondary antibodies (1/5,000) in blocking buffer for 1 h at room temperature. Immunoreactivity was detected with an enhanced chemiluminescence method (Western lightning® Plus-ECL, Perkin Elmer, Ref NEL103E001EA) on a ChemiDoc™ Touch Imaging System (Bio-Rad). Quantification was performed using the Image Lab software (Bio-Rad). Uncropped blots corresponding to the blots illustrated on the figures are provided in the Source Data file.

**Immunohistochemistry**. Eight-week-old mice were anesthetized with pentobarbital (100 mg/kg i.p., Ceva SA) and perfused transcardiacally with fixative solution containing 4% w/v paraformaldehyde, 0.1 M sodium phosphate buffer (pH 7.5), NaF (10 mM), and Na$^+$-vanadate (2 mM). Brains were post-fixed for 48 h in the same solution. For cryoprotection, tissue was transferred to PBS buffer containing 10 then 20 and finally 30% sucrose. They were then embedded in the embedding medium for cryotomy (Optimal Cutting Temperature, OCT, VWR Chemicals, Ref 361603E) and rapidly frozen with SnapFrost® (Excilone) and stored at −80 °C. 50 μm-thick coronal cryosections were cut using a Leica CM3050 Cryostat and kept free floating in PBS at 4 °C. Slices were incubated in PBS solution containing 10% heat-inactivated goat serum (Vector Laboratories, Ref S-100) and 0.3% Triton X-100 for 20 min. They were then incubated overnight at 4 °C in PBS containing 3% heat-inactivated goat serum, 0.1% Triton X-100, anti-Cx43 (Rabbit), anti-GFP (Chicken) and anti-GFAP (Mouse) antibodies. After extensive PBS washings, slices were incubated for 2 h with the Alexa Fluor® 488-conjugated anti-chicken antibody (1/1,000, Thermo Fisher Scientific, Ref A-11039), the Alexa Fluor® 594-conjugated anti-rabbit antibody (1/1,000, Thermo Fisher Scientific Thermo Fisher Scientific, Ref A-11037) and the Alexa Fluor® 680-conjugated anti-mouse antibody (1/1,000, Thermo Fisher Scientific, A-21057) in PBS containing 3% heat-inactivated goat serum, 0.1% Triton X-100 and Hoechst 33342 (2 μM, Thermo-Scientific, Ref 62249). After three washes in PBS, slices were mounted on superfrost ultra plus slides (Thermo-Scientific, Ref 10417002) using fluorescent mounting medium (Dako, Ref S3023). Pictures were taken with a Leica SP8-UV confocal microscope. Quantification of the Cx43 gap junctional plaque size was performed using the Fiji software. Each image was converted into binary and threshold was automatically defined. Plaque areas were then calculated by means of the Particle Analysis Fiji algorithm. The quantified areas were then divided by the number of cells counted in the field.

**Immunocytochemistry**. Astrocytes cultivated on coverslips were fixed with a 4% solution of PFA in PBS for 10 min. Glioblastoma cell lines cultured as free-floating cell spheres were mechanically dissociated, transferred onto SuperFrost glass slides (Dutscher) and fixed with 4% PFA in PBS. Excess of PFA was quenched by washing cells three times in a 0.1 M solution of glycine in PBS. Cells were permeabilized with a PBS solution containing 5% heat-inactivated goat serum and 0.1% Triton X-100 for 20 min. When needed, antigen retrieval was performed by heating up cells to 80 °C for 20 min in a citrate buffer solution (pH 6) containing Tween 0.05% before any staining. After cooling down to room temperature, cells were washed three times and incubated with primary antibodies (anti-Cx43 (rabbit or mouse), anti-ACKR3 (mouse) and anti-CXCR4 (rabbit)) in PBS containing 2.5%

heat-inactivated goat serum and 0.05% Triton X-100. After three washes in PBS, cells were incubated for 2 h at room temperature in PBS containing 2.5% goat serum, 0.05% Triton X-100, the appropriate secondary antibody (Alexa Fluor® 488-conjugated anti-mouse antibody (1/1,000, Thermo-Scientific, Ref A-11029), Alexa Fluor® 594-conjugated anti-rabbit antibody (1/1,000)) and Hoechst 33342 (2 μM). Cells were then washed three times with PBS and coverslips were mounted on superfrost ultra plus using fluorescent mounting medium. Pictures were taken at different confocal plans with a Leica SP8-UV confocal microscope. For the internalization assay the different confocal plans were then merged using the Z project algorithm of Fiji with the maximal intensity for projection. Plaque size was quantified using the counting particle plug-in of Fiji. For co-localization studies, 3D reconstructions from imaged confocal plans (between 20 and 50 plans for each cells) were performed using the Imaris software (Bitplane). Overlapping volume was firstly defined by identifying the center of mass of each object belonging to group A (Cx43) or B (ACKR3/CXCR4). Then, two objects, defined by setting an automatic threshold[64], were considered as co-localized if the center of one of them falls into the area of the other[65]. The intensity of Cx43 signals within the overlapping volume was quantified. For comparison this intensity was divided by the total intensity of each group before plotting.

**Time-lapse imaging of Cx43-GFP in living cells**. Transfected HEK293T cells were seeded (50,000 cells/well) in polyornitine-coated 35 mm-Nunc™ Glass Bottom Dishes (Thermo-Scientific, Ref 150680). Eighteen hours before the experiments, the medium was replaced by DMEM without serum. Cells were then washed with PBS and incubated in medium containing 130 mM NaCl, 2.8 mM KCl, 1 mM CaCl₂, 2 mM MgCl₂ and 10 mM HEPES (pH 7.2) before challenge. Cells were next treated with either CXCL12 (10 nM) or CXCL11 (100 nM) and imaged on a Leica SP8-UV inverted confocal microscope using a 488-nm argon/krypton laser line and a 561-nm DPSS laser line with a 63x oil (1.4 numerical aperture) objective. Cells were maintained at 37 °C, 95% O₂/ 5% CO₂ for the duration of the experiment during which the focus, contrast, or brightness settings remained constant. Optical scans were collected continuously at a scan speed of 32 s for 30 min after the treatment onset. For analysis, images were organized sequentially in a movie sequence using the LAS X software (v. 3.5, Leica).

**BRET**. Transfected HEK293T cells were seeded (50,000 cells/well) in white and black polyornithine-coated-96-well plates (Greiner, Ref 655083 and 655079). After 24 h, cells were washed twice with PBS. Coelenterazine (Nanolight Technology, Ref 50909-86-9) was added at a final concentration of 5 μM in the white plate before incubation at 37 °C for 5–10 min. Reading was immediately performed after the addition of different ligands or PBS, using a Mithras LB 940 plate reader (Berthold Biotechnologies) that allows the sequential integration of light signals detected with two filter settings (Rluc/NLuc filter, 485 ± 20 nm; and YFP filter, 530 ± 25 nm). Data were collected using the MicroWin2000 software (Berthold Biotechnologies). The YFP signal was measured from the black plate using an INFINITE F500 plate reader (TECAN) by setting the excitation filter at 485 ± 20 nm and the emission filter at 520 ± 10 nm.

**DERET**. DERET internalization assay was performed as described in[66]. Briefly, transfected HEK293T cells were seeded in white polyornithine-coated 96-well plates (25,000 cells/well). Forty-eight h after seeding, culture medium was substituted with 75 μl Tag-lite® labelling medium (Cisbio, Ref LABMED) containing 100 nM SNAP-Lumi4-Tb (Cisbio, Ref SSNPTBX). Cells were then incubated for 1.5 h at 16 °C. After washing the excess of SNAP-Lumi4-Tb, internalization experiments were carried out by incubating cells in the absence or presence of incremental concentrations of CXCL12 in Tag-lite® labeling medium containing fluorescein (25 μM, Merck, Ref 46955) at 37 °C for 1.5 h. Lumi4-Tb was excited at 340 nm and signals emitted at 620 and 520 nm were collected using the INFINITE F500 plate reader (TECAN). Ratio (620/520) was obtained by dividing the donor signal by the TR-FRET signal and multiplying this value by 10,000. Data were expressed as % of maximal internalization after subtraction of the internalization at the onset of treatment.

**Scrape loading**. Five-week-old astrocytes were serum-starved overnight in the presence or absence of PTX (100 ng/mL) or CBX (50 μM). Cultures were exposed to the indicated treatment in medium containing 130 mM NaCl, 2.8 mM KCl, 1 mM CaCl₂, 2 mM MgCl₂ and 10 mM HEPES (pH 7.2), for 30 min or 5 min at room temperature. Cells were then exposed to the same medium without CaCl₂ for 1 min and the cell monolayer was scraped using a razor blade in the presence of Lucifer yellow (1 mg/mL) dissolved in the calcium-free medium. LY was then let diffusing into cells for 1 min. Cells were then washed 5 times with the calcium-containing medium. After 10 min, images were taken using an inverted fluorescence microscope (Zeiss Axiovert 40CFL) equipped with a CCD camera (Axiocam ICCL1, Zeiss), using the Axiovision 4.8 software (Zeiss). For quantification of LY diffusion from the scrape, images were analyzed with the Fiji software. Fluorescence was plotted against the distance from the cut. The average fluorescence of the 50 most distant pixels from the cut was considered as background and subtracted from all values. Fluorescence was then normalized to the maximal fluorescence value and one exponential decay regression was fitted using Prism (v. 8.0, GraphPad Software

Inc.). Distances from the scrape where the fluorescence was 50% of the maximal fluorescence were considered for LY diffusion comparison between the experimental conditions after normalization to 100% in vehicle-treated astrocytes.

**Double patch-clamp recordings**. All experiments were performed in the whole-cell configuration of the patch-clamp technique. Junctional currents (Ij) were recorded in astrocyte cell pairs from secondary cultures using the dual voltage-clamp technique[67]. Coverslips were transferred to a recording chamber attached to the stage of an upright microscope (Axioskop FS; Zeiss) and continuously superfused with Ringer's saline (in mM): 125 NaCl, 2.5 KCl, 2 $CaCl_2$, 1 $MgCl_2$, 1.25 $NaH_2PO_4$, 26 $NaHCO_3$, 12 glucose and buffered to pH 7.4. The saline was maintained at 32 °C and continuously bubbled with carbogen (95% $O_2$/5% $CO_2$). Patch pipettes were pulled to a resistance of 4–5 MΩ from borosilicate glass (1.5 mm outer diameter; 1.17 mm inner diameter) and filled with an internal solution (in mM): 140 potassium-gluconate, 2 $MgCl_2$, 1.1 EGTA, 5 HEPES, and titrated to pH 7.2 with KOH. Junctional currents were acquired with an EPC-9 dual patch-clamp amplifier (HEKA Electronik) in cell pairs voltage-clamped at −50 mV and were filtered at 3 kHz[68]. In each cell pair, the stimulated cell was challenged with depolarizing voltage steps (40 mV amplitude, 300 ms duration, one step every 30 s) and the magnitude of the resulting current in the non-stimulated cell (junctional current Ij) was continuously monitored for 15–20 min. CBX- or chemokine-containing solutions were bath-applied through the perfusion system at a rate of 2 ml/min. Control cell pairs were challenged with saline, using the same protocol.

**Quantification of cAMP production**. Transfected HEK293T cells were seeded in 24-well plates (100,000 cells/well). Forty-eight h later, they were starved of serum overnight in the presence or absence of PTX (100 ng/mL). Confluent astrocyte cultures were starved of serum overnight in the presence or absence of PTX (100 nM). When indicated, cells were pre-exposed for 30 min to AMD3100. They were then stimulated for 10 min with CXCL11 or CXCL12 (in DMEM containing 1% BSA). Forskolin (1 µM, Merck, Ref F6886) was added 5 min after the treatment onset in the presence of 1 mM of the phosphodiesterase inhibitor 3-isobutyl-1-methylxanthine (Merck, Ref I5879). Cells were then lysed in 1% Triton X-100 for 30 min and cAMP production was quantified using the cAMP dynamic kit (Cisbio Bioassays, Ref 62AM4PEC) according to the manufacturer's instructions.

**Statistics and reproducibility**. Statistical analysis of the ACKR3 interactome (Fig. 1a) was performed using the Perseus software (v 1.5.6.072). Proteins were considered statistically significant using a t-test by setting the randomization number at 250, the False Discovery Rate at 0.01 and the S0[69] at 0.1. All other statistical analyses were performed using Prism (v.8.0, GraphPad Software Inc) and the statistical tests used are indicated in each legend. The statistical tests performed and the resulting P values are depicted on Supplementary Table 2. Dose-response curves (Supplementary Fig. 4b, d) were fitted by the log (agonist) vs. response (four parameters) non-linear regression using Prism. Supplementary Fig. 3d was obtained by fitting the log (antagonist) vs. response (three parameters) non-linear regression. Saturation BRET experiments were analyzed by using Prism and comparing, via an F-test, one-site total line with background constraint to 0 fitting and linear fitting through origin. All data are presented as means ± SEM. Significance levels were defined as $p < 0.05$ (*/$), $p < 0.01$ (**/$$), $p < 0.001$ (***/$$$), and $p < 0.0001$ (****/$$$$), where * represents statistical significance compared to the control and $ represents statistical significance between highlighted groups.

The interactomic screen illustrated on Fig. 1a was repeated in three biologically independent replicates. The immunohistochemistry experiments represented on Fig. 1b were conducted on three different animals with similar results. HA-immunoprecipitation on Fig. 1c was performed in three independent experiments performed from different mice with similar results. BRET signals on Fig. 1d were from five independent experiments performed on different sets of cultured cells. HA-immunoprecipitation on Fig. 1e was repeated in three independent experiments performed on different sets of cultured cells. Immunoreactive signals reconstructed on Fig. 1f were from 5 (Cx43 and ACKR3 in TG1 and R633 cells), 4 (Cx43 and CXCR4 in TG1 cells) or 7 (Cx43 and CXCR4 in R633 cells) cells. Kinetic BRET on Fig. 1g was measured in four independent experiments performed on different sets of cultured cells. HA-immunoprecipitation on Fig. 1h were repeated in five independent experiments performed on different sets of cultured cells. Scrape loading represented in Fig. 2 were conducted in three independent experiments performed in duplicate from different sets of cultured cells. Double patch-clamp recordings showed on Fig. 3 were from six cell pairs recorded in independent experiments performed on different sets of cultured cells. IF staining depicted in Fig. 4a was performed and measured in three different sets of cultured cells. Scrape loading represented in Fig. 4b were conducted in three independent experiments performed in duplicate from different sets of cultured cells. The time-lapse experiments of Fig. 5 were conducted independently in three different sets of cultured cells with similar results (see Supplementary Fig. 5).

**Reporting summary**. Further information on research design is available in the Nature Research Reporting Summary linked to this article.

## Data availability

Source data are provided with this paper (see Source Data File). The mass spectrometry data have been deposited to the ProteomeXchange Consortium via the PRIDE[70] partner repository with the dataset identifier PXD016978.

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

## Acknowledgements

The authors thank all our colleagues from the ONCORNET consortium for continuous scientific discussions and support, Prof. Joël Bockaert (IGF) for critical reading of the manuscript and Steeve Thirard (iExplore facility, RAM/Biocampus) for technical assistance with animals. Time-lapse recordings and imaging experiments were performed using facilities of the MRI platform of the Biocampus service unit. LC-MS/MS analyses were performed using the facilities of the Functional Proteomics Platform of the Biocampus service unit. BRET and DERET experiments were performed using the facilities of the Arpege pharmacological screening platform (Biocampus). This research was funded by a European Union's Horizon2020 MSCA Program [Grant agreement 641833 (ONCORNET)]. A.F., S.C.D., and P.M. are also supported by CNRS, INSERM, Université de Montpellier and Fondation pour la Recherche Médicale (FRM).

## Author contributions

A.F. conceived, performed, analyzed most of the experiments and wrote the manuscript; J.H. performed the BRET and DERET experiments; A.P. performed the time-lapse experiments and contributed to the co-immunoprecipitation; E.M. contributed to the generation and analysis of BRET data; J.K. performed some IHC experiments; M.S. generated and analyzed LC-MS/MS data; T.D. contributed to the design and analysis of experiments and manuscript revision; M.P.J. generated the glioma cell lines, designed immunocytochemistry experiments on these cells and revised the manuscript; G.SL. and F.B. designed IHC experiments and revised the manuscript; D.S. performed co-immunoprecipitation from embryonic mice brains; R.S. provided EGFP-ACKR3 BAC mice and HA-ACKR3 mice, designed experiments on these mice and revised the manuscript; M.J.S. participated in the project design and funding and revised the manuscript; N.C.G. performed the dual patch experiments, contributed to the analysis of the data and to the writing of the manuscript; S.C.D. performed molecular biology experiments and revised the manuscript; P.M. conceived and supervised the project, and wrote the manuscript.

## Competing interests

The authors declare no competing interests.
