## [Peer Review File · Nature Communications]

Reviewers' comments:

Reviewer #1 (Remarks to the Author):

This paper, "ACKR3 activation promotes Connexin 43 internalization to inhibit gap junctional intercellular communication" by Fumagalli et al. investigates the association and functional relationship of ACKR3 with Cx43, two proteins which have been reported to be involved in regulation of tumor proliferation. The authors first identified this potential interaction following characterization of the interactome of Cx43. Combining biochemistry and immunolabeling with gap junction functional studies the authors provide evidence supporting a role of ACKR3 in downregulating gap junctional communication following activation of these receptors by specific agonists. Further, the authors provide evidence that this inhibitory role is mediated by beta-arrestin2 and dynamin-dependent internalization of Cx43 channels. The evidence is overall convincing and the paper is clearly written and illustrated. There are however a number of concerns:

- The most important concern, and I admit that this is entirely subjective, is what makes these findings highly significant to deserve publication in a high-impact journal? The functional relationship between two tumor proliferation proteins was somehow expected. The study might potentially be highly significant but the authors didn't persuade me this is the case.
- Beyond the exhaustive and convincing evidence for the interaction between these two molecules, the paper is missing evidence of this functional interaction during tumor proliferation. The addition of data showing the impact of the described interactions in some type of model of tumor proliferation would enormously improve the paper, as well the overall impact of these findings.

Other comments:

- Page 6: "Previous studies have identified the C-terminal domain of GPCRs as the major site for interaction with their protein partners³⁰. The amount of Cx43 co-immunoprecipitated with HAACKR3 expressed in HEK293T cells was not affected by the partial or complete truncation of ACKR3 C-terminal domain (Supplementary Fig. 2), ruling out a role of this domain in Cx43 recruitment". I don't understand the need of reporting these experiments, as the authors do not provide the alternative site of interaction.
- Page 7: "...cells engaged in a gap junctional electrical coupling exhibited a significantly lower membrane resistance" What do the authors mean with "membrane resistance"? It seems the authors are rather measuring the "input resistance" of the cells.
- Page 8: "By contrast, inhibition of Cx43-mediated electrical cell-cell communication by CBX (100 μ M) significantly increased cell resistance (from 54.3 ± 5.2 to 306 ± 52.6 M Ω , $p = 0.002$, $n = 10$), indicating that the membrane resistance value reliably reflects a change in gap junction-mediated electrical coupling between two cells". This seems to be a circular argument.
- Page 8: "...on the electrical junctional communication between astrocytes, the amplitude of the junctional current". I suggest replacing "amplitude" for "magnitude".

- Page 8: I don't understand the need of expressing conductance as a ratio. Coupling coefficients are used when recording voltage changes in current clamp experiments, as they are not only influenced by the junctional conductance but also the input resistance of the cells. Under voltage clamp configuration, measurements of I_j will suffice.

- Page 8 and Figure 3b: "The coupling ratio remained stable during the 15-min recording of vehicle-treated cells, but..., and remaining inhibited as long as 10 min after the onset of treatments". What happened after 10 minutes? Why were the recordings so short?

- Page 9: "...suggesting that the ability of ACKR3 to activate Gai/o proteins depends on the cellular background rather than different coupling properties of ACKR3 originating from different species". I don't think the authors provide sufficient evidence to reach this conclusion.

Reviewer #2 (Remarks to the Author):

This is a highly interesting and well performed and documented study which provides compelling evidence that ligand-dependent activation of CXCR7 inhibits astrocytic gap junction coupling by promoting internalization of Cx43. This interaction has been previously unknown and might be crucial to various pathological brain processes. Despite these merits, the study would further benefit from tying up some of the loose ends and by providing some additional specific information:

1. Did the authors test whether the CXCL12-ACKR3 axis affects gap junction coupling / Cx43-internalization in their glioma cell lines? This demonstration would further underline the authors' claim of a potential pathophysiological role of the CXCL12-ACKR3-Cx43 "pathway" in gliomas.
2. Since CXCL11 was only tested in few experiments, it remains unclear whether the effects observed following CXCL12-dependent activation of ACKR7 fully compare to those induced by CXCL11? This issue is especially interesting since a previous study claimed that unlike CXCL12 CXCL11-bound CXCR7 would not activate G proteins, but β -arrestin 2 in astrocytes (Glia. 2012 Mar;60(3):372-81).
3. I further wonder if CXCL12/ACKR3-induced internalization of Cx43 is associated with (secondary) effects on Cx30? In this respect, immunocytochemistry would at least allow to assess putative effects on the subcellular localization of this connexin isoform.
4. It would definitely broaden the interest of the manuscript if the authors would (briefly) discuss or at least give some hints on the putative role of the CXCR7-Cx43 pathway in the diseased brain (in addition to gliomas).
5. I miss mentioning for which experiments primary and secondary astrocyte cultures have been used. This information is important since primary and secondary cultures might differ with respect to their cellular composition. Likewise, I miss mentioning of the specific age of the animals used for visualizing co-localisation of ACKR3 and Cx43 (Fig. 1b) and for immunoprecipitation (Fig. 1c).

6. Page 12, lines 3-5; I guess this sentence needs to be reworded. The sentence is only correct if the authors mean that ACKR3 and Cx43 have opposite effects on a given cellular process. However, if the authors mean that findings on the function of ACKR3 and Cx43 are conflicting, in terms that depending on the study either promoting or inhibiting effects were observed, the sentence is misleading.

Reviewer #3 (Remarks to the Author):

This manuscript uses a pulldown approach to identify molecular ACKR3 interactors in HEK293 cells and then examines functional aspects of one of them, the gap junction protein Cx43. Novelty and interest are fairly high given potential implications for glioma biology. However, a major concern is that data are not so compelling for certain claims in the manuscript. In particular internalization is not demonstrated, BRET data are not unambiguous, differences between AACKR3 and CXCR4 colocalization are not convincing, Cx43 isoform intensities differ in input and IP lanes, coupling ratio rather than junctional conductance is quantified.

Specific comments:

Title: Abbreviation should be changed to either a general statement of what ACKR3 is; demonstration of internalization is not strong.

P2, I2 and elsewhere: Change “different” to various or multiple or diverse

L4. Expand this sentence to state that pulldown was with exogenous tagged protein and that HEK293 cells were used

L12. This sentence is confusing...so the authors mean that it is the first demonstration of functional interaction or the first such demonstration with pathological relevance?

P3, I4. In the last...

P4, I3. Cx43 channels are involved in these functions, not Cx43 itself. What may be most relevant to the present study is the modulation of Cx43 function through interactions with cytoplasmic proteins, of which there are many examples.

L10. That under some condition the expression of one protein is increased while function of another is decreased is weak evidence for “a possible negative interplay”.

L14. Change drastically to substantially or greatly or another term not suggesting sudden or disruptive action

L17. “Common influence” seems a strange way to describe what was referred to on the previous page as negative interplay.

P5,I7. If I understand the Methods correctly, these were consistent hits in all three biological replicates; if so, that should be stated here. There appear to be a few proteins with significantly less interaction with the HAOagarose beads than in the sham; how is this interpreted? Are they interesting proteins?

L21. But the screen did not pick up the Cx43 binding partners that other pulldowns have identified, such as α -tubulin, catenins and ZO-1.

p.6, I8. Did the authors compare pulldown results from brain with those from HEK293 cells with

respect to other potential binding partners?

L13. Perhaps a statement should be added regarding the significance of hyperbolic binding.

L22. The amount of Cx43 colPeped after agonist stimulation is only incrementally higher

Last line. The complete truncation seems less effective [$\sim 50\%$ increase], perhaps not reaching statistical significance due to low n and high variance.

Fig.1a: What is the interpretation of proteins with negative numbers? What is the rationale for FDR=1%?

Fig 1b: In mouse brain, what is the % overlap of staining?

Fig1c and others (1e, 1g): Cx43 input in many cases only shows a single band, whereas IP shows double bands. Assuming that the upper band corresponds to the major protein in Input, the presence of substantial lower band (generally referred to as dephosphorylated in the connexin field) may suggest that the interaction with ACKR3 is primarily with the de-/nonphosphorylated form. This form is thought to be the major Cx43 present in non-plaque domains.

Fig 1d: What is the evidence that the ACKR3 curve is hyperbolic and the CXCR4 curve is not? Why is maximum BRET signal in CXCR4 only 80%? The maximal difference between BRET values in the two curves is about 20%, which is approximately the deficit in peak BRET % for CXCR4. In the bottom histogram, the difference may be significant but it is not substantial ($\sim 10\%$), and this should be mentioned in results.

Fig 1f. The substantially lower ($\sim 80-90\%$ lower) overlapping volumes for CXCR4 than ACKR3 are not evident in the images above the histograms.

Fig 1g. While the 30 min treatment with CXCL12 was significantly higher than before treatment, the 5 min treatment was not. This differs from what is stated in the text.

Fig 2. Images of PTX treated samples all (except CBX) show much more extensive dye passage than in the vehicle groups. Why is that?

Fig. 3 and supplemental data showing lower input resistance in coupled than uncoupled cells: The "coupling ratio" (which is generally used when coupling is measured under current clamp conditions) is not the correct parameter, since it depends on nonjunctional as well as junctional conductance. What should be plotted is junctional conductance ($-I_2/V$). Information that input resistance is lower in coupled cells is consistent with this, but is not relevant to the effects of the drugs on gap junctions.

Fig. 4a. Cell size and/or confluence is very different in the various panels (eg, # nuclei is second panel WT is at least twice that in the first panel and cells are quite small. Puncta seem quite variable and quantitation of % plaque size is obscure, perhaps accounting for large variability in the Dynasore dataset (where $>50\%$ difference is not significant). The conclusion from this set of experiments is that Cx43 internalization is stimulated by CXCL11 and 12, which is blocked by Dynasore and in the beta arrestin KO mouse. However, there is no measure comparing the baselines (which are used to normalize plaque data) and there is no evidence of GJ internalization rather than dispersion or even channel closing.

Responses to the Reviewers' comments:

Reviewer #1:

This paper, "ACKR3 activation promotes Connexin 43 internalization to inhibit gap junctional intercellular communication" by Fumagalli et al. investigates the association and functional relationship of ACKR3 with Cx43, two proteins which have been reported to be involved in regulation of tumor proliferation. The authors first identified this potential interaction following characterization of the interactome of Cx43. Combining biochemistry and immunolabeling with gap junction functional studies the authors provide evidence supporting a role of ACKR3 in downregulating gap junctional communication following activation of these receptors by specific agonists. Further, the authors provide evidence that this inhibitory role is mediated by beta-arrestin2 and dynamin-dependent internalization of Cx43 channels. The evidence is overall convincing and the paper is clearly written and illustrated.

There are however a number of concerns:

- The most important concern, and I admit that this is entirely subjective, is what makes these findings highly significant to deserve publication in a high-impact journal? The functional relationship between two tumor proliferation proteins was somehow expected. The study might potentially be highly significant but the authors didn't persuade me this is the case.

The functional relationship between two tumor proliferation proteins is not obvious, as they might operate through distinct and entirely independent mechanisms. It is therefore important to establish if such a relationship exists, as it can contribute to or even amplify the cellular effects of the two proteins. The current study not only provides the first evidence of a physical and functional link between ACKR3 and the gap junction protein Cx43, but also unveils a new mode of regulation of gap junctional intercellular communication by a GPCR that involves the internalization of gap junction proteins through a β -arrestin-dependent mechanism. Furthermore, as outlined in the discussion (page 12, paragraph 2), the negative interplay between ACKR3 and Cx43 GJIC might underlie some key pathophysiological effects of CXCL12 and CXCL11, including neuronal migration, leukocyte entry into the brain, inflammatory demyelinating disorders and cancer progression. We feel that these are important findings that might be of interest for a broad readership working in various research fields such as GPCRs, gap junctional intercellular communication, brain diseases and cancer and that they deserve publication in a high-impact journal such as Nature Communications.

- Beyond the exhaustive and convincing evidence for the interaction between these two molecules, the paper is missing evidence of this functional interaction during tumor proliferation. The addition of data showing the impact of the described interactions in some type of model of tumor proliferation would enormously improve the paper, as well the overall impact of these findings.

We agree with the reviewer's assertion that the role of the functional interaction between ACKR3 and Cx43 in tumor proliferation is an important issue that certainly warrants further exploration. We are currently planning studies to address that point. Please note that the editors indicated in the decision letter that "they will not require further insights into the role of the ACKR3-Cx43 pathway in glioma proliferation as they consider this beyond the scope of the current manuscript"

Other comments:

- Page 6: "Previous studies have identified the C-terminal domain of GPCRs as the major site for interaction with their protein partners³⁰. The amount of Cx43 co-immunoprecipitated with HAACKR3 expressed in HEK293T cells was not affected by the partial or complete truncation of ACKR3 C-terminal domain (Supplementary Fig. 2), ruling out a role of this domain in Cx43 recruitment". I don't understand the need of reporting these experiments, as the authors do not provide the alternative site of interaction.

We agree with the reviewer that this is not essential information and we therefore removed these data from the manuscript. We only kept the following statement in the discussion: "The role of the physical ACKR3-Cx43 interaction in receptor-operated inhibition of GJIC could not be fully established in absence of the identification of binding motifs within the sequences of both partners" (see page 13, lines 4-6) because we feel that this is important information.

- Page 7: "...cells engaged in a gap junctional electrical coupling exhibited a significantly lower membrane resistance" What do the authors mean with "membrane resistance"? It seems the authors are rather measuring the "input resistance" of the cells.

We agree with the reviewer and replaced "membrane resistance" by "input resistance", see pages 7 (last paragraph) and 8.

- Page 8: "By contrast, inhibition of Cx43-mediated electrical cell-cell communication by CBX (100 μ M) significantly increased cell resistance (from 54.3 ± 5.2 to 306 ± 52.6 M Ω , $p = 0.002$, $n = 10$), indicating that the membrane resistance value reliably reflects a change in gap junction-mediated electrical coupling between two cells". This seems to be a circular argument.

We modified this sentence and specified that the CBX effect confirms that the input resistance can be used as an index for change in gap junctional intercellular communication (see page 8, lines 2-5).

- Page 8: "...on the electrical junctional communication between astrocytes, the amplitude of the junctional current". I suggest replacing "amplitude" for "magnitude".

We replaced "amplitude" by "magnitude" as suggested (see page 8, paragraph 2).

- Page 8: I don't understand the need of expressing conductance as a ratio. Coupling coefficients are used when recording voltage changes in current clamp experiments, as they are not only influenced by the junctional conductance but also the input resistance of the cells. Under voltage clamp configuration, measurements of I_j will suffice.

We agree with the reviewer. Therefore, we have now plotted the I_j on Figures 3b-c. The text has also been changed accordingly (see page 8, paragraph 2).

- Page 8 and Figure 3b: "The coupling ratio remained stable during the 15-min recording of vehicle-treated cells, but..., and remaining inhibited as long as 10 min after the onset of treatments". What happened after 10 minutes? Why were the recordings so short?

Double patch-clamp recording of junctional currents in astrocyte cultures is highly challenging due to the difficulties to continuously record cell pairs over periods of time longer than 15 min (4 min before the onset of the treatment and 11 min after). Nevertheless, we successfully recorded the I_j of 2, 3 and 4 cell pairs from CXCL12-, CXCL11- and CBX-treated cultures, respectively, for at least 20 min (i.e. up to 16 min after the onset of chemokine or CBX application). To take into consideration the Reviewer's comment, we also plotted the I_j magnitude during the last five min of recording for those cells and showed that the I_j was still inhibited (see the Figure below). This suggests that ACKR3 stimulation induces a long-lasting inhibition of GJIC that persists at least for 15 min after the onset of receptor activation. As the number of cell pairs recorded beyond 15 min is limited, we prefer not to include these data in the manuscript. Unfortunately, we did not succeed in recording cell pairs beyond 20-25 min.

- Page 9: "...suggesting that the ability of ACKR3 to activate Gai/o proteins depends on the cellular background rather than different coupling properties of ACKR3 originating from different species". I don't think the authors provide sufficient evidence to reach this conclusion.

We agree with the reviewer's criticism and have modified this conclusion (see page 9, end of paragraph 2).

Reviewer #2

This is a highly interesting and well performed and documented study which provides compelling evidence that ligand-dependent activation of CXCR7 inhibits astrocytic gap junction coupling by promoting internalization of Cx43. This interaction has been previously unknown and might be crucial to various pathological brain processes. Despite these merits, the study would further benefit from tying up some of the loose ends and by providing some additional specific information:

We thank the reviewer for his/her positive comments.

1. Did the authors test whether the CXCL12-ACKR3 axis affects gap junction coupling / Cx43-internalization in their glioma cell lines? This demonstration would further underline the authors' claim of a potential pathophysiological role of the CXCL12-ACKR3-Cx43 "pathway" in gliomas.

To address the reviewer's query, we performed scrape loading experiments in glioma cell lines, such as U251 cells (De Ridder *et al.*, *Acta Neuropathol.*, 72:207-13, 1987). However, we found no GJIC in these cells (as assessed by Lucifer yellow diffusion), corroborating previous studies that show a low Cx43 expression and Lucifer yellow diffusion in this cell line (Zhang *et al.*, *J Exp Clin Cancer Res.*, 29:3, 2010; Gentry *et al.*, *Gene Therapy*, 12:1033–1041, 2005). Accordingly, we could not detect any additional inhibitory effect of ACKR3 stimulation in those glioma cells.

*2. Since CXCL11 was only tested in few experiments, it remains unclear whether the effects observed following CXCL12-dependent activation of ACKR7 fully compare to those induced by CXCL11? This issue is especially interesting since a previous study claimed that unlike CXCL12 CXCL11-bound CXCR7 would not activate G proteins, but β -arrestin 2 in astrocytes (*Glia*. 2012 Mar;60(3):372-81).*

Although CXCL11 was tested in the most representative experiments assessing ACKR3-mediated Cx43 inhibition (scrape loading, double patch-clamp recordings) and inhibition of cAMP production, data showing the effect of CXCL11 in most if not all functional readouts are now included in the manuscript. Especially, we provide BRET and co-immunoprecipitation results indicating that CXCL11 enhances the recruitment of Cx43 by ACKR3, thus reproducing the CXCL12 effects (see Figures 1g and h, and "Results", pages 6 (4 last lines) and 7.

3. I further wonder if CXCL12/ACKR3-induced internalization of Cx43 is associated with (secondary) effects on Cx30? In this respect, immunocytochemistry would at least allow to assess putative effects on the subcellular localization of this connexin isoform.

As requested by the reviewer, we performed immunocytochemistry experiments assessing Cx30 localization in astrocytes. The data (illustrated on Supplementary Figure 7 and described in "Results", pages 10, last paragraph, and 11) show a predominant intracellular localization of Cx30 in astrocytes that is not affected by CXCL12 or CXCL11 treatment. Exposure of cells to dynasore induced a redistribution of Cx30 to the cell surface in cells treated or not with CXCL11 and CXCL12. Collectively, these results suggest that Cx30 is "constitutively" internalized in astrocytes through a mechanism independent of ACKR3 stimulation and that ACKR3-induced internalization of Cx43 does not influence Cx30 trafficking in astrocytes.

4. It would definitively broaden the interest of the manuscript if the authors would (briefly) discuss or at least give some hints on the putative role of the CXCR7-Cx43 pathway in the diseased brain (in addition to gliomas).

As suggested by the reviewer, we now mention the role of the ACKR3-Cx43 pathway in multiple sclerosis and other inflammatory demyelinating diseases of the central nervous system, which

suggest that its pathological influence is not restricted to glioma (see “Discussion”, page 12, paragraph 2).

5. I miss mentioning for which experiments primary and secondary astrocyte cultures have been used. This information is important since primary and secondary cultures might differ with respect to their cellular composition. Likewise, I miss mentioning of the specific age of the animals used for visualizing co-localisation of ACKR3 and Cx43 (Fig. 1b) and for immunoprecipitation (Fig. 1c).

Primary cultures were used for all experiments on astrocytes except double patch-clamp recordings that were performed on secondary cultures. This is now clearly mentioned in the “Materials and Methods” section, page 18, end of paragraph 2 and throughout the “Results” section. We used secondary cultures for double patch-clamp recordings in order to obtain isolated pairs of astrocytes. The fact that the results of patch-clamp recordings in secondary cultures corroborate those obtained in scrape loading experiments performed in primary cultures certainly reinforce the strength of the data.

We apologize for omitting to indicate the age of the mice used in immunohistochemistry experiments illustrated on Figure 1b. This information has been added in the “Materials and Methods” section, page 21, paragraph 2 and in the figure legend. The age of animals used for immunoprecipitations illustrated on Figure 1c was indicated in the first version (see Page 19, paragraph 2).

6. Page 12, lines 3-5; I guess this sentence needs to be reworded. The sentence is only correct if the authors mean that ACKR3 and Cx43 have opposite effects on a given cellular process. However, if the authors mean that findings on the function of ACKR3 and Cx43 are conflicting, in terms that depending on the study either promoting or inhibiting effects were observed, the sentence is misleading.

As pointed by the reviewer, we mean that ACKR3 and Cx43 have opposite effects on common processes. There are not conflicting results but a consensus on these opposite effects. The sentence has been modified to take into consideration the reviewer’s comment (see page 12, paragraph 2, 3 first lines).

Reviewer #3

This manuscript uses a pulldown approach to identify molecular ACKR3 interactors in HEK293 cells and then examines functional aspects of one of them, the gap junction protein Cx43. Novelty and interest are fairly high given potential implications for glioma biology. However, a major concern is that data are not so compelling for certain claims in the manuscript. In particular internalization is not demonstrated, BRET data are not unambiguous, differences between AACKR3 and CXCR4 colocalization are not convincing, Cx43 isoform intensities differ in input and IP lanes, coupling ratio rather than junctional conductance is quantified.

Specific comments:

Title: Abbreviation should be changed to either a general statement of what ACKR3 is; demonstration of internalization is not strong.

We agree with the reviewer and have modified the title according to his comment. We propose as new title "The atypical chemokine receptor 3 interacts with Connexin 43 and inhibits gap junctional intercellular communication".

We also performed time-lapse recordings of Cx43 trafficking, which further support Cx43 internalization upon ACKR3 stimulation by CXCL11 or CXCL12 (see Figure 4b, Supplementary Figure 6, and our response to the last comment).

P2, I2 and elsewhere: Change "different" to various or multiple or diverse

We replaced "different" by "various" in the abstract and throughout the manuscript.

L4. Expand this sentence to state that pulldown was with exogenous tagged protein and that HEK293 cells were used

We expanded the sentence and specified that exogenous tagged ACKR3 was precipitated from HEK293T cells, as requested.

L12. This sentence is confusing...so the authors mean that it is the first demonstration of functional interaction or the first such demonstration with pathological relevance?

We agree with the reviewer that the original sentence was ambiguously written. We now clearly state that the present study provides the first demonstration of a functional link between ACKR3 and gap junctions.

P3, I4. In the last...

We do not understand what the reviewer is asking in this point. So, we did not modify the sentence.

P4, I3. Cx43 channels are involved in these functions, not Cx43 itself. What may be most relevant to the present study is the modulation of Cx43 function through interactions with cytoplasmic proteins, of which there are many examples.

We agree with the reviewer and the correction has been made (see page 4, lines 2-3).

L10. That under some condition the expression of one protein is increased while function of another is decreased is weak evidence for "a possible negative interplay".

We modified the text according to the reviewer's comment (see page 4, end of paragraph 1).

L14. Change drastically to substantially or greatly or another term not suggesting sudden or disruptive action

We replaced “drastically” by “substantially” (see page 4, line 15).

L17. “Common influence” seems a strange way to describe what was referred to on the previous page as negative interplay.

We agree with the reviewer’s comment and have modified the sentence accordingly (page 4, end of paragraph 2).

P5, 17. If I understand the Methods correctly, these were consistent hits in all three biological replicates; if so, that should be stated here.

Only proteins identified in all three biological replicates in at least one group were indeed considered for relative quantification and statistical analysis. This point, already mentioned in the “Materials and Methods” section, has now been included in “Results” (page 5, lines 9-10).

There appear to be a few proteins with significantly less interaction with the HA0agarose beads than in the sham; how is this interpreted? Are they interesting proteins?

We agree with the reviewer that finding proteins with significantly higher abundance in the control immunoprecipitation, compared to the ACKR3 IP, is counterintuitive. However, such a situation is often observed in large-scale interactomic studies (see, for instance, Oliviero *et al.*, *Mol Cell Proteomics*, 15:3450-3460, 2016; Schopp *et al.*, *Nat Commun*, 8:15690, 2017; Alsulami *et al.*, *Mol Cell Proteomics*, 18:1428-1436, 2019). In all cases, a minority of proteins was found more enriched in the control situation (three proteins in our interactomics screen) and this was implicitly attributed to chance (these proteins are considered as false hits). One explanation is that those three proteins have some affinity for the HA-antibody conjugated to the beads. Consequently, they would freely bind to the beads in the control condition, whereas they would compete with the HA-ACKR3 in the experimental conditions lowering their abundance. The three proteins found statistically enriched in the control condition in our screen are 1) Actin-related protein 8 (Q9H981, ACTR8), 2) Probable global transcription activator SNF2L1(P28370, SMARCA1) and 3) Mediator of RNA polymerase II transcription subunit 23 (Q9ULK4, MED23). As they are not potentially involved in the ACKR3 effects, they were not considered for further analyses.

L21. But the screen did not pick up the Cx43 binding partners that other pulldowns have identified, such as α -tubulin, catenins and ZO-1.

First of all, it is noteworthy that large-scale interactomic studies are never exhaustive. The failure to detect a protein in such screens does not mean that the protein is absent. In addition, ACKR3 was used as bait in our experiments, whereas Cx43 was co-immunoprecipitated with the receptor. Therefore, Cx43 binding partners would be indirectly co-precipitated within a complex formed by ACKR3, Cx43 and Cx43 binding partners. Accordingly, the failure to identify previously described CX43-interacting proteins might reflect stoichiometry issues. Furthermore, the instability of some protein-protein interactions makes the identification of indirectly recruited proteins often challenging. In spite of these limitations, we successfully identified proteins previously shown to be physically or functionally linked to Cx43 and to regulate Cx43-mediated GJIC (Dynactin 1, Desmosomal cadherin desmoglein 2, Ubiquillin-4, Cytochrome P450 oxidoreductase, Solute carrier family 1 member 5 and the beta-subunit of the electron-transfer protein). These proteins were already highlighted in Figure 1 and mentioned in the original text (see page 5, last paragraph).

In addition, we checked again whether the Cx43 partners mentioned by the reviewer are present in the list of ACKR3 interacting proteins identified in our screen. ZO-1 and catenins were not

identified. However, we identified tubulin beta-3 chain (TUBB3) that together with alpha-tubulin (Giepmans *et al.*, Cell Commun Adhes., 8:219-23, 2001) interacts with Cx43, as a protein significantly enriched in our ACKR3 interactome.

We have modified Figure 1 and the related text (page 6, lines 2-4) in order to add this protein and we apologize for missing it in the previous version.

p.6, l8. Did the authors compare pulldown results from brain with those from HEK293 cells with respect to other potential binding partners?

We agree with the reviewer that comparing pulldown results from brain with those from HEK293 cells with respect to other potential binding partners showing a great enrichment in ACKR3 precipitates would be of great interest. In this regard, we confirmed the association of ACKR3 with G α i3 by BRET (see Supplementary Figure 3b), even though this experiment could only be performed in HEK293T cells. The identification of a number of proteins known to be involved in ACKR3 trafficking or signal transduction (see page 5, lines 14-21) further validates our interactomics screen. We decided to focus on ACKR3 interaction with Cx43 rather than further explore the interaction of the receptor with other candidate partners, because both proteins were shown to influence several common pathophysiological processes.

L13. Perhaps a statement should be added regarding the significance of hyperbolic binding.

We have added a statement explaining the significance of hyperbolic binding (see page 6, line 16).

L22. The amount of Cx43 coIPed after agonist stimulation is only incrementally higher. Last line. The complete truncation seems less effective [~50% increase], perhaps not reaching statistical significance due to low n and high variance.

We agree with the reviewer regarding the incremental increase of Cx43 co-immunoprecipitation upon agonist stimulation of the receptor. This has been specified in the text (see page 6, lines 28-29).

As suggested by Reviewer#1, we removed the experiments using truncated ACKR3, because they did not provide any extra information. Furthermore, they were inconclusive, as they did not reach statistical significance.

Fig.1a: What is the interpretation of proteins with negative numbers? What is the rationale for FDR=1%?

The False Discovery Rate (FDR) corresponds to the expected proportion of false positives in a list of putative candidates. It is a recommended parameter in high-throughput experiments with potentially many differentially expressed proteins. A 5% FDR is proposed as default setting by Perseus, but we used 1% FDR (stringent criterion) to limit the number of false positives in our study.

Fig 1b: In mouse brain, what is the % overlap of staining?

As reported in the legend to Figure 1 and in the "Materials and Methods" section, *Ackr3*-EGFP BAC mice were used for immunohistochemistry experiments. These mice express EGFP under the promoter of *Ackr3* inserted into a random location of the genome. Though ACKR3 expressing cells will also express eGFP, a cytosolic protein, both proteins exhibit different sub-cellular localizations. Consequently, the % of overlap of Cx43 and eGFP staining did not reflect the actual colocalization of ACKR3 and Cx43, and was not calculated.

Fig1c and others (1e, 1g): Cx43 input in many cases only shows a single band, whereas IP shows double bands. Assuming that the upper band corresponds to the major protein in Input, the presence of substantial lower band (generally referred to as dephosphorylated in the connexin field) may suggest that the interaction with ACKR3 is primarily with the de-/nonphosphorylated form. This form is thought to be the major Cx43 present in non-plaque domains.

Only one immunoreactive band corresponding to Cx43 was detected in the input from the mouse brain extract (Figure 1c), corroborating the results of Fiorini *et al.* (Fiorini *et al.*, J Cell Sci. 117:4665-72, 2004) that showed a predominant expression of one Cx43 isoform (likely the dephosphorylated one) in the brain. In contrast, multiple immunoreactive bands are detected in the inputs from HEK293T cells (Figures 1 e and h). These include a major band at around 40 kDa, one band of lower apparent molecular weight and several bands of higher molecular weights. The major band and the lower molecular weight band correspond to dephosphorylated Cx43, as they are not affected by the addition (4 h, 30°C) of calf intestinal phosphatase to the cell lysate, in contrast to the upper bands which disappear after incubation of the lysate with the phosphatase (see the figure below).

Only the dephosphorylated Cx43 forms, especially the lower molecular weight form, were co-immunoprecipitated with the receptor. To further explore whether this low molecular weight form corresponds to an internally-translated Cx43 isoform (Smyth *et al.*, Cell Reports, 5:611-8, 2013), HEK293T cells were treated with rapamycin (20 ng/ml, 2 h at 37°C) in order to prevent CAP-dependent translation of proteins and to promote the translation of the shorter Cx43 isoform. However, the treatment did not change the proportion of the different immunoreactive bands, indicating that the lower molecular weight band does not correspond to internally-translated Cx43. This form remains to be characterized. As it was not detected in the brain, we feel that these results are not essential and would prefer not to include them in the manuscript. Of course, if the reviewer and the editors consider they are important data, we will incorporate them.

Fig 1d: What is the evidence that the ACKR3 curve is hyperbolic and the CXCR4 curve is not? Why is maximum BRET signal in CXCR4 only 80%? The maximal difference between BRET values in the two curves is about 20%, which is approximately the deficit in peak BRET % for CXCR4. In the bottom histogram, the difference may be significant but it is not substantial (~10%), and this should be mentioned in results.

As stated in the legend to Figure 1d “For each dataset (ACKR3, CXCR4) the best-fitting model between a one-site-total-line with background constraint to 0 and a line through origin was plotted” using the Prism software. For the ACKR3 dataset, Prism fitted the one-site-total-line (P value < 0.001, $R^2=0.97$) as the best model. The same type of fitting for the CXCR4 dataset showed that a line through the origin ($R^2=0.94$) is the best model.

The BRET signals depicted on Figure 1d were normalized to the maximal BRET signal obtained in cells co-expressing ACKR3-Nluc and Cx43-YFP. As the BRET signal in cells expressing CXCR4-Nluc increased linearly with Cx43-YFP, we could not calculate a maximal BRET.

Note that we replaced the histogram illustrated in the previous Figure 1d by a graph showing normalized BRET as a function of the incubation time with either CXCL12 and CXCL11 (added in

order to respond to Reviewer #1's query). This new graph highlights the time-dependency of the association of ACKR3 with Cx43 upon receptor stimulation by either CXCL12 or CXCL11 (see new Figure 1g). As stated by the reviewer, both treatments induced a slight but significant increase in BRET signals. Note that such "modest" differences in BRET are frequently observed, as illustrated, for instance, in Supplementary Figure 3e assessing changes in BRET signal between $G\alpha_{i3}$ -RLuc and Venus- $\gamma 2$ following activation of CXCR4 by CXCL12, a classical test measuring the dissociation of the heterotrimeric G protein as an index of its activation. Again, the maximal BRET variation was inferior to 10%.

Fig 1f. The substantially lower (~80-90% lower) overlapping volumes for CXCR4 than ACKR3 are not evident in the images above the histograms.

We agree with the reviewer and replaced the picture illustrating CXCR4 immunostaining on Figure 1f by a more representative one.

Fig 1g. While the 30 min treatment with CXCL12 was significantly higher than before treatment, the 5 min treatment was not. This differs from what is stated in the text.

We have changed Figure 1g (now Figure 1h) to show the effect of CXCL11 upon the co-immunoprecipitation Cx43 with ACKR3, as requested by Reviewer #1. We have also suppressed the 5-min treatment from the figure, as it did not significantly change the association of Cx43 with ACKR3. We have also modified the text accordingly and specified that the co-immunoprecipitation of Cx43 with ACKR3 was incrementally increased upon ACKR3 stimulation by CXCL12 or CXCL11 (see page 6, 2 lines from bottom).

Fig 2. Images of PTX treated samples all (except CBX) show much more extensive dye passage than in the vehicle groups. Why is that?

We agree with the reviewer that PTX treatment increases the dye diffusion through the astrocyte syncytium, as already pointed in the "Results" section. This might reflect the activation of endogenously expressed $G_{i/o}$ -coupled receptors that inhibit Cx43-mediated GJIC as a result of their activation by gliotransmitters released by astrocytes or constitutive activity. This point has been added in the discussion (see page 13, paragraph 2).

Fig. 3 and supplemental data showing lower input resistance in coupled than uncoupled cells: The "coupling ratio" (which is generally used when coupling is measured under current clamp conditions) is not the correct parameter, since it depends on nonjunctional as well as junctional conductance. What should be plotted is junctional conductance ($-I_2/V$). Information that input resistance is lower in coupled cells is consistent with this, but is not relevant to the effects of the drugs on gap junctions.

We agree with the reviewer's criticism (also raised by Reviewer #1) and now report the normalized I_j instead of the coupling ratio in Figure 3 and Supplementary Figure 2. Values are identical to those obtained in plotting the junctional conductance, as the experiments were performed at constant voltage (clamped at -50 mV).

There are contradictory results indicating that connexins can influence (Recabal *et al.*, Front Cell Neurosci, 12:406, 2018) or not (Pannasch *et al.*, Proc Natl Acad Sci U S A., 108:8467-72, 2011) astrocyte input resistance. Our data obtained in CBX-treated cells indicate that uncoupled cells have a higher resistance. This suggests that the input resistance can be used as an indirect index for change in GJIC and reinforces the data obtained by measuring the I_j .

Fig. 4a. Cell size and/or confluence is very different in the various panels (eg, # nuclei is second panel WT is at least twice that in the first panel and cells are quite small. Puncta seem quite

variable and quantitation of % plaque size is obscure, perhaps accounting for large variability in the Dynasore dataset (where >50% difference is not significant). The conclusion from this set of experiments is that Cx43 internalization is stimulated by CXCL11 and 12, which is blocked by Dynasore and in the beta arrestin KO mouse. However, there is no measure comparing the baselines (which are used to normalize plaque data) and there is no evidence of GJ internalization rather than dispersion or even channel closing.

As noticed by the Reviewer, the cell and nuclei sizes in astrocyte cultures are rather heterogenous. To address the reviewer's concern, we have changed some of the pictures of the initial figure to illustrate fields with more homogenous cell or nucleus sizes. We have also normalized all the data to the value obtained in untreated WT astrocytes, to compare the different baselines. These results initially depicted on Figure 4a, are now illustrated on a new supplementary figure (Supplementary Figure 5).

Most importantly, to get more direct evidence that ACKR3 stimulation promotes Cx43 internalization, we performed time-lapse recordings of Cx43-GFP in living HEK-293T cells. We show that exposing cells co-expressing ACKR3 and Cx43-GFP to CXCL12 or CXCL11 induces the removal of Cx43 plaques localized at cell-cell contacts, whereas plaques are not affected in cells that do not express ACKR3. This likely reflects Cx43 internalization, as Gap junction buds were transiently observed prior to the removal of plaques. Furthermore, removal of Cx43 plaques often started from their central part, consistent with previous findings indicating that Cx43 internalization primarily takes place from the center of the plaques (Gaietta *et al.*, *Science*, 296:503-507, 2002). Galleries illustrating the effects of CXCL12 or CXCL11 upon Cx43 trafficking are now illustrated on Figure 4b and Supplementary Figure 6, and described in "Results", page 10, paragraph 2. A new paragraph describing the protocol used for time-lapse recordings of Cx43-GFP in living cells has also been added in the "Materials and Methods" section (see page 23, paragraph 2).

REVIEWER COMMENTS

Reviewer #1 (Remarks to the Author):

The authors have substantially improved the paper in response to the reviewers' comments. All my technical concerns/comments have been addressed. However, while the judgment on the significance of the other two reviewers persuaded me, the authors should make more efforts in explaining the general significance of the findings. Tumor proliferation involves multiple proteins, so it is not surprising to find interactions between two of them. Moreover, there is no data supporting the interaction occurs during tumorigenesis. Again, a stronger, more general, statement on the significance of the findings will enormously improve the readability of the paper.

Reviewer #2 (Remarks to the Author):

The authors adequately addressed all issues brought up. I have no further problems with the manuscript.

Reviewer #3 (Remarks to the Author):

The authors have made major revisions to this manuscript that greatly improve both clarity and conclusions. Although many of my comments were addressed, a few concerns remain.

My previous comment regarding P3, I4 was omitted from the review (but was included in my previous concern about comparing the pulldown results from brain with those from HEK293 cells) involved the issue of the cell type expression of ACKR3. It is stated that it is found in neurons, astrocytes and vascular cells, and the lack of total overlap with GFAP staining in micrographs is consistent with this. The study cited to support use of embryonic brain reported highest expression in neurons and vessel wall, although some GLAST+ expression was noted, presumably in astrocytes. Neurons do not express Cx43, and therefore most cells expressing ACKR3 do not express it. This should be addressed in the manuscript.

Pulldown of desmosomal cadherin desmoglein 2 is somewhat problematic and should not be highlighted as evidence for relevance of identified binding partners. The reference cited to indicate previous report was in cardiac myocytes, which certainly express desmosome components. But expression in astrocytes is highly unlikely.

My concern remains regarding the interaction of ACKR3 primarily with the de-/nonphosphorylated form of Cx43. It should be noted that this form is not thought to be the major Cx43 present in GJ plaques but may be in non-plaque domains. Interestingly perhaps, such domains might include retrieved intracellular vesicles.

I remain concerned about the conclusion that the authors strongly attribute a difference in mechanism of binding (saturated receptor vs random interaction) solely based on the highest R2 values for fittings. What is relevant is whether the fittings of the two curves to the same model are statistically different or not, not the highest r2 values.

While it is true that the higher input resistance after CBX uncoupling is consistent with decreased coupling, this need not be the case (as when one or both cells become leaky), and it is incorrect to state that "...the input resistance can be used as an indirect index for change in GJIC and reinforces the data obtained by measuring the I_j "

Reviewer #4 (Remarks to the Author):

The manuscript by Fumagalli describes the interaction of the atypical chemokine receptor 3 (ACKR3) with Connexin 43 and the functional interplay of these proteins in mouse brain astrocytes and human glioblastoma cells. This ultimately results in altered gap junctional intercellular communication.

The study is based on a MS-based interactomics screen which was performed by forced expression of ACKR3 in HEK293T cells. Such interactomics studies of transmembrane receptors are still highly challenging and this is especially true when efficient antibodies are lacking. An overexpression study as has been performed by the authors in this manuscript is a valid alternative, but only provided sufficient validation of selected interactions is included. The quality of the proteomics study is high which is clearly supported by a number of proteins in the list of candidate partners that have been described before. The likelihood that other interactions in the list will be biologically relevant is good, but this requires further validation.

From the list of candidate partners, based on different criteria, the authors selected Cx43, which they have extensively validated by an impressive array of experiments: overexpression Co-IP, endogenous Co-IP in mouse embryonic brain, orthogonal assay (BRET), co-localization, by specific association to this receptor and not a related one, and by stimulus-dependent increase of the association.

As a minor detail it would be best to also report the S_0 value that was used to determine the cut-off in the volcano plot in Figure 1a (in addition to the FDR).

RESPONSES TO THE REVIEWER'S COMMENTS

Reviewer #1:

The authors have substantially improved the paper in response to the reviewers' comments. All my technical concerns/comments have been addressed. However, while the judgment on the significance of the other two reviewers persuaded me, the authors should make more efforts in explaining the general significance of the findings. Tumor proliferation involves multiple proteins, so it is not surprising to find interactions between two of them. Moreover, there is no data supporting the interaction occurs during tumorigenesis. Again, a stronger, more general, statement on the significance of the findings will enormously improve the readability of the paper.

To address the reviewer's concern, we have added in the "Conclusions" paragraph (page 15, lines 8-21) a general statement outlining the importance of characterizing protein-protein interactions involving molecules upregulated and mobilized in a diversity of cancers, such as ACKR3, to understand the mechanisms underlying cancer progression and develop new therapeutic strategies.

To further support the significance of the findings, we have also added a sentence pointing the key role of ACKR3 interacting proteins in the functional outcomes of this receptor revealed by our study at the end of the introduction (page 4, 3 last lines).

Reviewer #3

The authors have made major revisions to this manuscript that greatly improve both clarity and conclusions. Although many of my comments were addressed, a few concerns remain.

My previous comment regarding P3, I4 was omitted from the review

The correction has been made in the new version of the manuscript (see page 3, line 4).

(but was included in my previous concern about comparing the pulldown results from brain with those from HEK293 cells) involved the issue of the cell type expression of ACKR3. It is stated that it is found in neurons, astrocytes and vascular cells, and the lack of total overlap with GFAP staining in micrographs is consistent with this. The study cited to support use of embryonic brain reported highest expression in neurons and vessel wall, although some GLAST+ expression was noted, presumably in astrocytes. Neurons do not express Cx43, and therefore most cells expressing ACKR3 do not express it. This should be addressed in the manuscript.

We agree with the reviewer's comment regarding the partial overlap of ACKR3 and Cx43 immunostainings in the brain, consistent with the fact that neurons express ACKR3, but not Cx43. Accordingly, the interaction of ACKR3 with Cx43 only occurs in restricted cell populations in the brain. This point is now addressed in the discussion (page 12, lines 13-17).

Pulldown of desmosomal cadherin desmoglein 2 is somewhat problematic and should not be highlighted as evidence for relevance of identified binding partners. The reference cited to indicate previous report was in cardiac myocytes, which certainly express desmosome components. But expression in astrocytes is highly unlikely.

Our interactomic screen was performed in HEK293T cells that express desmosomal cadherin desmoglein 2 (Inada *et al.*, International Journal of Molecular Medicine 37: 1521-1527, 2016). Therefore, the association of ACKR3 with both Cx43 and desmoglein 2 in this cellular background is plausible. However, we agree with the reviewer about the fact that expression of desmosomal cadherin desmoglein 2 is unlikely in astrocytes. Accordingly, we removed the highlight (previously on page 5, line 25) on desmosomal cadherin desmoglein 2 from the revised version.

My concern remains regarding the interaction of ACKR3 primarily with the de-/nonphosphorylated form of Cx43. It should be noted that this form is not thought to be the major Cx43 present in GJ plaques but may be in non-plaque domains. Interestingly perhaps, such domains might include retrieved intracellular vesicles.

As pointed out by the reviewer, several lines of evidence suggest that the phosphorylated forms of Cx43 are major components of GJ plaques (reviewed in Solan *et al.*, Biochimica et Biophysica Acta (BBA) – Biomembranes, 1: 83-90, 2018). However, contradictory results also showed that the integrity of the carboxyl-terminal domain of Cx43, which contains the majority of Cx43 phosphorylated sites so far described, is not necessary for the formation of gap junction plaques, thus arguing against the notion that Cx43 phosphorylation is mandatory to the formation of functional gap junction plaques (Maass *et al.*, Circulation Research 101(12):1283-91, 2007).

We previously mentioned in our response after the first revision that Cx43 from mouse brain and HEK293T cells exhibited different electrophoretic patterns: one immunoreactive band for brain Cx43 that likely corresponds to the dephosphorylated protein, vs. multiple immunoreactive bands that remain to be fully characterized in HEK293T cells. In addition, although phosphatase inhibitors were added in the lysis buffer, we cannot entirely rule out the possibility of Cx43 dephosphorylation during the co-immunoprecipitation procedure. Accordingly, whether ACKR3 preferentially binds to non-phosphorylated vs. phosphorylated forms of Cx43 remains uncertain. In view of these considerations and the contradictory published data regarding the phosphorylation state of Cx43 engaged in gap junction plaques, we prefer that Cx43 phosphorylation is not discussed in the current manuscript, as proposed in the previous revision.

I remain concerned about the conclusion that the authors strongly attribute a difference in mechanism of binding (saturated receptor vs random interaction) solely based on the highest R2 values for fittings. What is relevant is whether the fittings of the two curves to the same model are statistically different or not, not the highest r2 values.

We agree with the reviewer, the R2 value for fittings is not a relevant index to define the mechanism of binding. In fact, we did perform what was asked by the reviewer and apologize if it was not clear in our previous response. As previously stated in the legend to Figure 1d, “For each dataset (ACKR3, CXCR4) the best-fitting model between a one-site-total-line with background constraint to 0 and a line through origin was plotted” using the Prism software. For the ACKR3 dataset, Prism fitted the one-site-total-line as the best model (P value < 0.001), whereas a line through the origin was selected as the best fitting for the CXCR4 dataset. No P value was calculated for the CXCR4 dataset since the one-site-total line only ambiguously fitted the data. We added this information in the legend to Figure 1d (page 35, lines 13-17) to clarify that point.

We also wish to stress that BRET results indicating a specific interaction between ACKR3 and Cx43, in comparison of CXCR4, corroborate those obtained using different approaches (co-immunoprecipitation and immunofluorescence staining of glioma initiating cells).

While it is true that the higher input resistance after CBX uncoupling is consistent with decreased coupling, this need not be the case (as when one or both cells become leaky), and it is incorrect to state that "...the input resistance can be used as an indirect index for change in GJIC and reinforces the data obtained by measuring the I_j"

We agree with the reviewer and this statement (previously on page 8, lines 4-5) has been removed from the revised manuscript.

Reviewer #4:

As a minor detail it would be best to also report the S0 value that was used to determine the cut-off in the volcano plot in Figure 1a (in addition to the FDR).

The S0 value has been added in the graph (see Figure 1a).

REVIEWERS' COMMENTS:

Reviewer #3 (Remarks to the Author):

There are two unaddressed concerns in this revised manuscript::

- 1) The conclusion that the arbitrarily selected and statistically ambiguous linear fitting of BRET between CXCR4 and Cx43 “reflecting random collision between..” strongly overstates the data, which show only statistically nonlinear fitting of the Cx43-ACK BRET signal. The phrase newly inserted on p35 regarding implication does not correct this incorrect conclusion.
- 2) The dual whole cell patch clamp method unambiguously determines junctional conductance, the parameter of interest. Input resistance is irrelevant and changes are obvious from the recordings shown. Supplemental figure provides no useful information.

In addition, the statement that Cx43 has been viewed as a tumor suppressor (where refs 21 and 47 are cited in different sections of the manuscript are too simple a summary of what is a complex and contradictory literature, in which Cx43 may be required for metastasis etc. The consensus view currently is that the role of Cx43 in tumorigenesis may be context-specific.

Responses to Reviewer #3's remarks:

There are two unaddressed concerns in this revised manuscript:

1) The conclusion that the arbitrarily selected and statistically ambiguous linear fitting of BRET between CXCR4 and Cx43 “reflecting random collision between..” strongly overstates the data, which show only statistically nonlinear fitting of the Cx43-ACK BRET signal. The phrase newly inserted on p35 regarding implication does not correct this incorrect conclusion.

We removed from the legend to Figure 1 (page 37) the sentence regarding the fitting of the BRET signal between CXCR4 and Cx43 (“No P value was calculated for the CXCR4 dataset since the one-site-total line only ambiguously fitted the data.”) that the reviewer considered as an overinterpretation of the data. Furthermore, we modified the text on page 6 in the “Results” section to only focus on the non-linear fitting of the ACKR3-Cx43 BRET signal suggestive of a specific interaction between both proteins in living cells (see paragraph 2, lines 12-16 in the Word document and 138-142 in the related pdf). We also slightly modified Figure 1d and now only illustrate the one-site saturation curve that fitted the ACKR3/Cx43 dataset.

2) The dual whole cell patch clamp method unambiguously determines junctional conductance, the parameter of interest. Input resistance is irrelevant and changes are obvious from the recordings shown. Supplemental figure provides no useful information.

Although we feel that changes in input resistance are important information, we removed Supplementary Figures 2b and c and the corresponding text from the “Results” and “Materials and Methods” sections as well as the legend to Supplementary Figure 2, as suggested by the reviewer.

In addition, the statement that Cx43 has been viewed as a tumor suppressor (where refs 21 and 47 are cited in different sections of the manuscript are too simple a summary of what is a complex and contradictory literature, in which Cx43 may be required for metastasis etc. The consensus view currently is that the role of Cx43 in tumorigenesis may be context-specific.

We agree with the reviewer's comment and have tempered our statement regarding the role of Cx43 in tumorigenesis. We modified the Introduction accordingly and point the opposing roles of Cx43 in glioma progression (see page 4, lines 8-14 in the Word document and 80-86 in the related pdf). We also suppressed the term “opposite” a few lines later (see line 21 in the Word document and 93 in the related pdf). Likewise, we indicate in the Discussion that Cx43 can either act as tumor suppressor or oncogene, depending on the cancer type and stage (see pages 12, lines 18-19 in the Word document and 318-319 in the related pdf, and pag 15; lines 10-11 in the Word document and 397-398 in the related pdf).